# Adhesion strength, cell packing density and cell surface buckling in pericellular matrix-mediated tissue cohesion

Rudolf Winklbauer‡, Olivia Luu, Debanjan Barua*, Martina Nagel and Yunyun Huang

## ABSTRACT

Pericellular matrix-mediated cell-cell adhesion in *Xenopus* gastrula tissues is characterized by a spectrum of narrow and wide cell contacts that alternate with the non-adhesive surfaces of the interstitial space. Here we show, first, that knockdown of a pericellular matrix adhesion molecule, fibronectin, diminishes contact abundance, and hence cell-packing density, without reducing adhesion strength. Second, we find that cell surfaces in gastrula tissues exhibit solid-like behavior in the form of buckling and crumpling, shape modifications that are typically seen in thin elastic films. We propose that both phenomena are explained by generic properties of the pericellular matrix: its compression and consequent stiffening by the interpenetration of matrix layers during adhesive contact formation. We argue that this renders part of the cell surface non-adhesive to form interstitial gaps, and both gap surfaces and contacts prone to buckling and crumpling in line with cell contractility fluctuations. In this elasto-capillary model of tissue cohesion, the size of the interstitial space is determined by the abundance of the pericellular matrix, not by adhesion strength.

KEY WORDS: Cell adhesion, Cell packing, Pericellular matrix, Buckling, Crumpling, Fibronectin

## INTRODUCTION

In non-epithelial tissues held together by cell-cell adhesion, contacts between cells are often interspersed with gaps. The relative size of this fluid-filled interstitial space – or inversely, the cell-packing density – is an important characteristic of morphogenetical active tissues such as chick, mouse or zebrafish mesoderm (Granholm and Baker, 1970; Batten and Haar, 1979; Mongera et al., 2018; Huljev et al., 2023), zebrafish blastoderm (Morita et al., 2017), mouse blastocyst (Dumortier et al., 2019), limb bud mesenchyme (Thorogood and Hinchliffe, 1975; Damon et al., 2008), or developing mammary gland (Ewald et al., 2012). In *Xenopus*, the whole gastrula is permeated by interstitial channels (Barua et al., 2021), and the mammalian adult neocortex shows an interstitial reticulum of similar size and structure (Tonnesen et al., 2023).

In such biphasic tissues consisting of cellular and extracellular space, the interstitial gaps facilitate the transport of molecules by advection and diffusion (Levick, 1987; Recho et al., 2019; Tonnesen et al., 2023). In morphogenesis, they play important mechanical roles (Campas et al., 2024). In zebrafish axis elongation, transitions in the size of interstitial spaces regulate tissue fluidity (Mongera et al., 2018). Spatial differences in cell-packing density control doming in the zebrafish gastrula (Morita et al., 2017; Petridou et al., 2019), and reconfigurations of the interstitial space facilitate zebrafish mesoderm migration (Huljev et al., 2023) and drive blastocoel formation in the mouse blastocyst (Dumortier et al., 2019).

Despite the importance of interstitial gaps, their mechanics is only beginning to be understood. Intuitively, gaps could result from incomplete cell-cell adhesion (Parent et al., 2017). Presently, the paradigm for adhesion in tissues is the adherens junction of epithelia, where cadherins and other transmembrane molecules interact extracellularly while linked to the cortical cytoskeleton inside (Harris and Tepass, 2010; Troyanovsky, 2023). This adhesion mechanism is also proposed for non-epithelial tissues (Campas et al., 2024). It is usually combined with a capillarity principle – the notion that cell surfaces can be treated like fluid surfaces – and various types of tissue models have as essential ingredients surface tension analogs (Graner and Glazier, 1992; Winklbauer, 2015; Alt et al., 2017; Mongera et al., 2018; Dumortier et al., 2019).

Specifically, contractile tension of the actomyosin cortex minimizes free cell surface (Evans and Yeung, 1989; Salbreux et al., 2012), while lowering tension at contacts expands the contact area. Tension reduction is achieved by contact-induced cortex downregulation and/or by the release of binding energy upon adhesion molecule interaction. A directly related measure of adhesion strength is the tension difference between free surfaces and contacts – the tissue surface tension (Steinberg, 1978; Graner, 1993; Brodland and Chen, 2000; Manning et al., 2010; Maître et al., 2012; David et al., 2014; Winklbauer, 2015; Parent et al., 2017, 2024). In a respective model of biphasic *Xenopus* gastrula tissue, tension differences would promote cell-on-cell spreading until equilibrium is reached. When this would occur before cell-cell attachment is complete, interstitial gaps would remain (David et al., 2014; Barua et al., 2017; Parent et al., 2017). In such capillarity-based models, interstitial space decreases with increasing adhesion strength (Parent et al., 2017; Kim et al., 2021).

In *Xenopus* gastrula tissues, knockdown of adhesion factors reduces cell-packing density and increases gap size, but, surprisingly, adhesion strength at remaining contacts often seems undiminished (Barua et al., 2017, 2021; Barua and Winklbauer, 2022; Luu et al., 2024). Thus, the common capillarity-based models of cell-cell adhesion may not predict gap size. This recommends a re-examination of the adhesion mechanism, with special regard to biphasic interstitium-containing tissues (Parent et al., 2017; Barua et al., 2017; Recho et al., 2019; Dumortier et al., 2019; Kim et al., 2021; Huljev et al., 2023).

Department of Cell and Systems Biology, University of Toronto, Toronto M5S 3G5, Canada.
*Present address: Department of Quantitative Biosciences, Merck Research Laboratories, South San Francisco, CA 94080, USA

‡Author for correspondence (r.winklbauer@utoronto.ca)

R.W., 0000-0002-0628-0897; Y.H., 0000-0001-6488-5088

A revision of concepts is also prompted by the fact that the cell coat, or pericellular matrix (PCM), is involved in adhesion (Winklbauer, 2019). Challenging the view that adhesion is generally based on adherens-type junctions, many non-epithelial tissues have the narrow ~30 nm contacts expected for cadherins (Tepass et al., 2000) replaced by much wider contacts (e.g. Steinberg, 1962, 1963; Thorogood and Hinchcliffe, 1975; Singley and Solursh, 1980; Damon et al., 2008), and adhesion strengths are much higher than those possible with cadherins (Winklbauer, 2019). In the *Xenopus* gastrula, adhesion is essentially mediated by PCM (Barua et al., 2021; Luu et al., 2024). Membrane-membrane distances at cell contacts range from 10 to 1000 nm and are mostly incompatible with cadherin binding. Further, contacts are sensitive to the depletion of matrix components such as fibronectin (FN) or hyaluronic acid, and many contacts show a glycocalyx-type ultrastructure (Barua et al., 2021). The PCMs of two cells can interpenetrate and this seems to drive adhesion (Luu et al., 2024). Cadherins are nevertheless important. They trigger the downregulation of cortical tension at contacts (David et al., 2014) but also regulate the PCM ultrastructure (Barua et al., 2021; Luu et al., 2024). The mechanism of PCM adhesion is currently poorly understood, but it appears to differ fundamentally from cadherin adhesion.

We study here the intertwined processes of PCM-mediated adhesion and interstitial gap formation in *Xenopus* prechordal mesoderm. Using experimental depletion of the adhesion factor FN, we confirm by direct measurements our previous tentative claims that gap size and cell-cell adhesion strength are independent. We suggest a mechanism of gap size control based on generic properties of the PCM and argue that, instead of its adhesiveness, the bulkiness and elastic compressibility of the PCM determine the formation of interstitial space. The mechanism entails a bending stiffness at cell surfaces; the wrinkling and crumpling at gaps and contacts support this observation and suggest an elasto-capillary model of adhesion.

## RESULTS

### FN knockdown reduces cell-packing density independently of cell-cell adhesion strength

In the gastrula, prechordal mesoderm (PM) moves animally on the neuroectoderm, whereas chordamesoderm involution and convergent extension shift the blastopore vegetally (Fig. 1A). A fibrillar or punctate FN matrix covers all cells (Fig. 1B,C) (Winklbauer, 1998). FN knockdown (Fig. 1D-F) (Davidson et al., 2006; Nagel and Winklbauer, 2018) diminishes fibrils and puncta, and inhibits involution. The PM still advances animally but cells are loosely packed (Fig. 1D). The fraction $c$ of the PM cell surface engaged in contacts is halved in morphants, consistent with a role for FN in cell-cell adhesion. The non-adherent cell surfaces line the interstitial space that appears in sections as irregular gaps (Barua et al., 2017) (Fig. 1G,H). The average side length $l$ of gaps (Fig. 1G) increases almost twofold upon FN depletion (Fig. 1I), and gap area per tissue section area increases fourfold (Fig. 1J), quadrupling the size of the interstitial space.

In a randomly packed tissue, the number of cells around a gap, i.e. the number ($n$) of gap sides, is a characteristic variable (Fig. S1A-C). Geometrically, with constant side length $l$, as, for example, in regular polygons, gap size depends on $n$ (Fig. S1A). Unexpectedly, however, side length $l$ itself also varies with $n$. It is smallest in two- and three-sided gaps and larger for $n>3$ in normal or FN-depleted PM (Fig. 1K,L). Notably, in FN morphants $l$ is increased at each $n$. This is expected when a basic side length is determined by a general FN- and, hence, PCM-dependent mechanism, and then modified by tissue geometry.

Intuitively, one would expect that loose cell packing indicated low adhesiveness, but this is not the case. Adhesion strength can be quantitated as tissue surface tension σ (Winklbauer, 2015). In contrast to measuring the dynamic rupture of cell-cell contacts, which includes energy dissipation (Maître et al., 2012; Kashef and Franz, 2015), σ estimates the reversible work of adhesion in cell aggregates at equilibrium as the difference in free surface energy per area, or tension, between contacts and non-attached surfaces. Axisymmetric Drop Shape Analysis (ADSA) derives σ from the shape a tissue explant assumes under gravity (Fig. S2) (David et al., 2014). In ectoderm, FN depletion does not significantly lower contact abundance (Barua et al., 2021) and σ is not reduced (Fig. 2A). However, σ is not diminished in PM either (Fig. 2B), despite the decrease in cell-packing density, and co-depletion of C-cadherin and Syndecan-4 diminishes both packing density (Barua et al., 2021) and σ (Fig. 2C). Apparently, the two parameters are independent, and FN can modulate contact abundance without affecting adhesion strength.

This result can be analyzed by using the general formalism of a biphasic capillarity-based model when considering for the moment that tension as surface free energy/area may include elastic tension, as made explicit below (see Eqns 5a,b and 6a,b). For stable gaps, tensions at the tissue surface and at gaps must be balanced, and tissue adhesion strength σ and local adhesion strength $\sigma_i$ at gaps are distinguished (Figs 2D-D″ and S3) (Parent et al., 2017). σ quantitates the difference between tension β at the tissue surface and a reduced tension $\beta_c$ at cell contacts, and the ratio of the tensions determines the contact angle θ between cells (Fig. 2D′):

$$\sigma = \beta - \beta_c \tag{1a}$$

$$\cos \theta = \frac{\beta_c}{\beta}. \tag{1b}$$

Local adhesion strength $\sigma_i$ is defined in analogy to σ; the ratio of tensions at gap-contact transitions, with $\beta_f$ indicating the tension at gap surfaces, determines contact angle $\theta_i$ at gaps (Fig. 2D″):

$$\sigma_i = \beta_f - \beta_c \tag{2a}$$

$$\cos \theta_i = \frac{\beta_c}{\beta_f}. \tag{2b}$$

A dimensionless relative adhesiveness can be defined as:

$$\alpha = \frac{\sigma_i}{\beta_f} = 1 - \cos \theta_i. \tag{2c}$$

To see that gap size can be independent of overall tissue adhesion strength σ, one may, first, consider a generalized interstitial space inserted between two adherent tissue parts. The same tension $\beta_c$ runs through contacts and, virtually, also through this gap (Fig. 2D-D″). Inserting a larger gap reduces contact abundance; however, to separate the tissue parts, the same reversible work per area is still required: at both gaps and contacts, $\beta_c$ returns to β and thus σ remains the same. This argument does not depend on gap shape (Fig. S3). Second, measuring σ, θ and $\theta_i$ fixes the remaining four parameters in Eqns 1a,b and 2a,b,c and allows us to calculate $\sigma_i$. This parameter quantitates local adhesion strength also in the elasto-capillary model (see supplementary Materials and Methods). FN knockdown does not significantly alter the measured values (Fig. 2E,F) and, thus, $\sigma_i$ also remains the same, i.e. FN affects adhesion strength averages neither globally nor locally at gaps. This contrasts with C-cadherin depletion, the main effect of which is a smaller downregulation of β at contacts and, therefore, a lower surface tension σ (David et al., 2014).

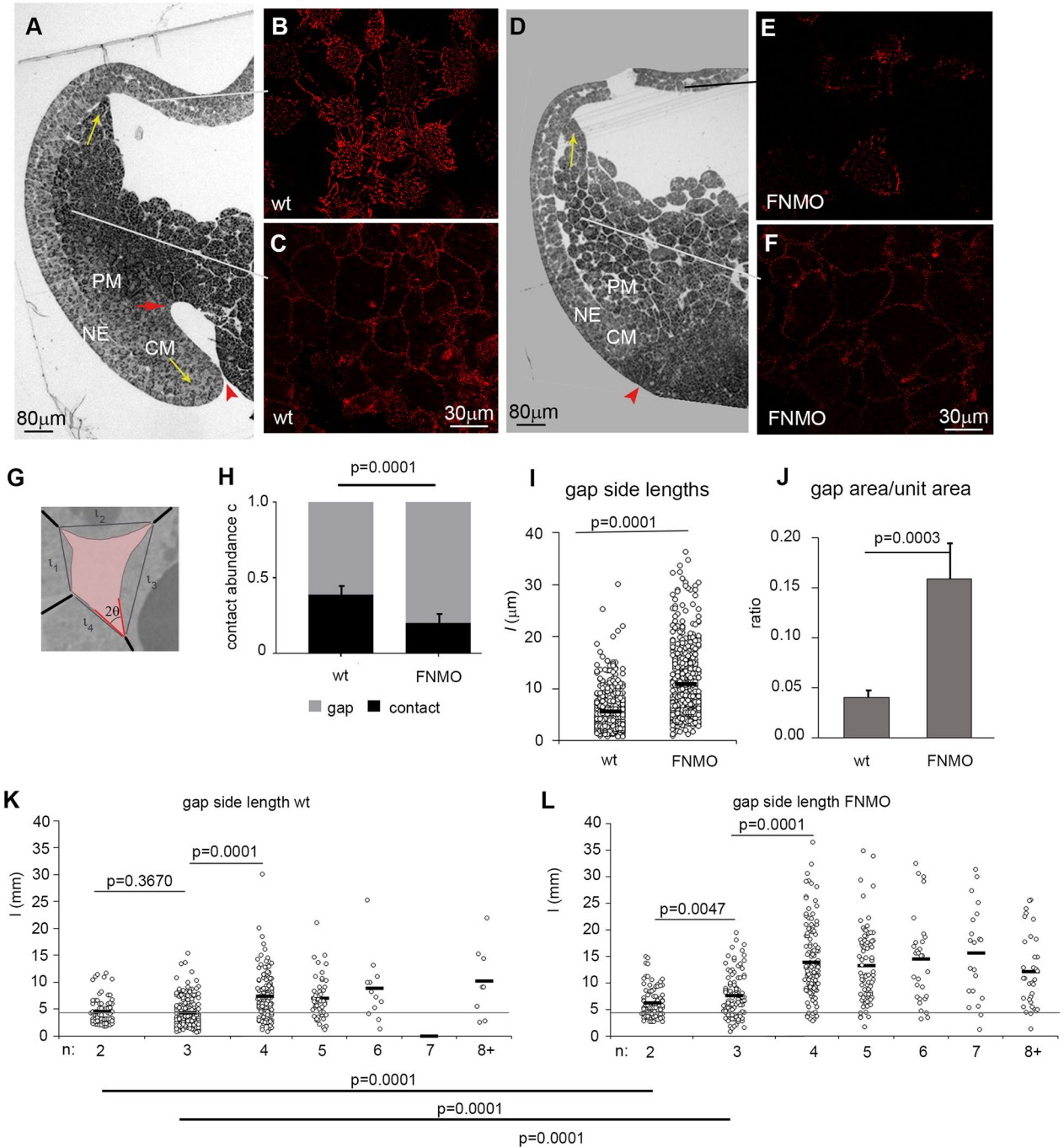

**Fig. 1. Cell-packing density and gap size.** (A) A stage 11 gastrula, dorsal side. NE, neuroectoderm; CM, chordamesoderm; PM, prechordal mesoderm. Yellow arrows indicate tissue movement; red arrowhead indicates blastopore; red arrow indicates archenteron tip. (B,C) Ectodermal surface (B) and deep PM cells (C) stained with FN antibody. (D) Stage 11 gastrula FN knockdown. Red arrowhead indicates the position of the blastopore in the absence of archenteron. (E,F) FN staining after knockdown. (B,C,E,F) Five specimens stained for each sample. (G) Interstitial gap (pink), side lengths ($l$), contact angle $2\theta_i$ and cell-cell contacts (bold black lines) are indicated. (H) Contact abundance as fraction, $c$, of total cell surface (black); the fraction of the remaining surface in gaps (gray), in normal (wild type, 24 embryos) and FN-depleted PM (FNMO, 12 embryos) (data from Barua et al., 2021). (I,J) Side lengths $l$ (I) and gap area (J) per tissue section area. (K,L) Gap side length ($l$) as function of gap side number ($n$). Gray horizontal reference line through the lowest average (at wild type $n$=2). (I-L) Measurements for wild type are from 24 TEM images, six embryos; measurements for FNMO are from 30 TEM images, three embryos. Averages are indicated by black horizontal bars; $P$ indicates significance levels (one-way ANOVA); vertical bars indicate s.d.

## Unevenly curved or kinked gap sides are incompatible with capillarity-based models

In capillarity-based models a defined and uniform curvature of gap surfaces is implied. For regular three-sided gaps, gap size $l$, contact angle $\theta_i$ and gap surface curvature $C_i$ are related by $l=(2/C_i)$ $\sin(30° - \theta_i)$ (Parent et al., 2017), with 30° being half the polygon angle $\theta_p$ (Fig. S1A). For $n$-sided gaps, $\theta_p$ increase regularly with $n$ (Fig. S1A) and $l=(2/C_i)$ $\sin((1-2/n)90° - \theta_i)$, i.e. curvature $C_i$ is fixed when $l$, $\theta_i$ and $n$ are given. However, PM gap surfaces differ decidedly from the expected evenly curved shapes, with contours

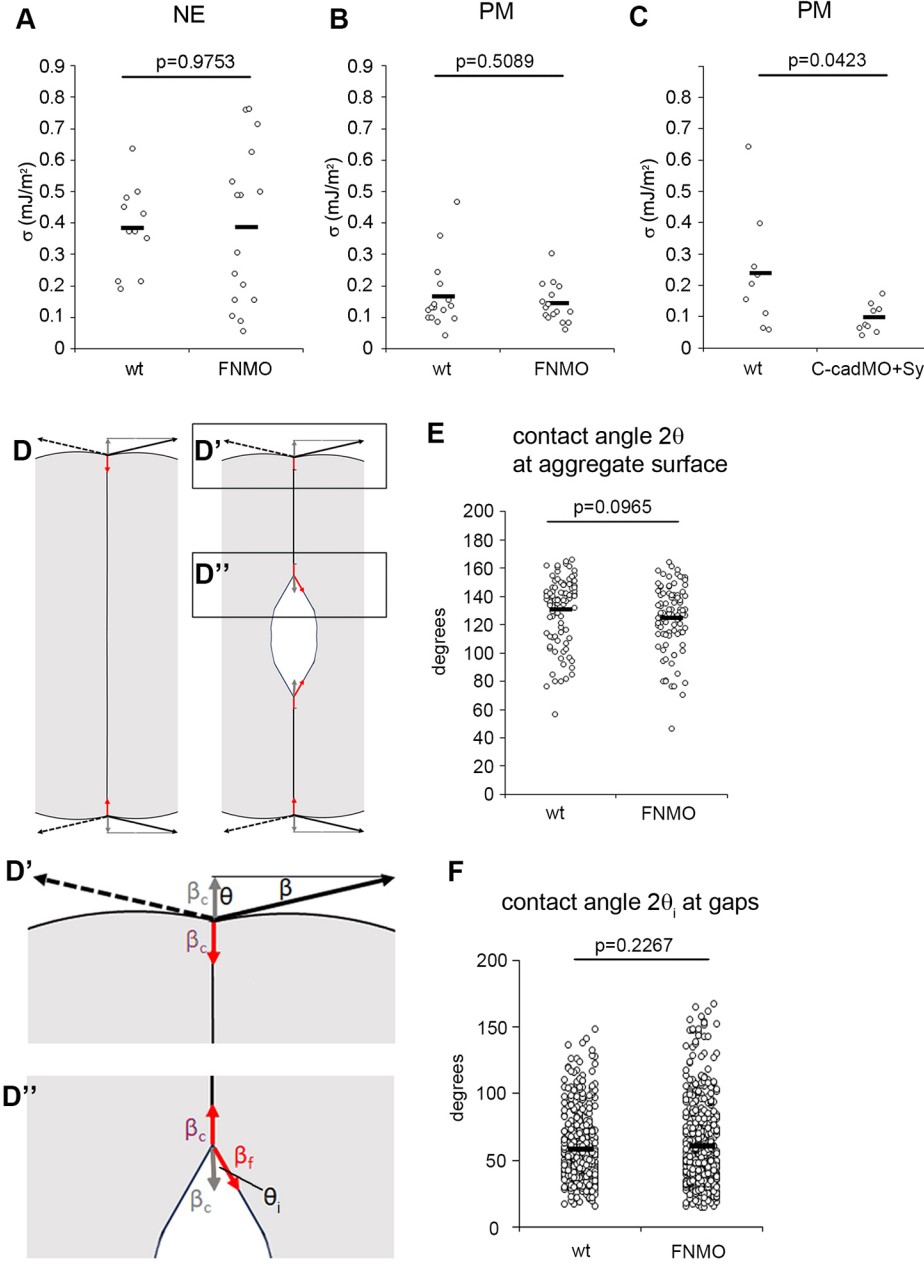

**Fig. 2. Tissue and cell surface tensions.** (A-C) Tissue surface tension measured by ADSA in normal and FN-depleted neural ectoderm (A), PM (B) or C-cadherin- and Syndecan-4-depleted PM (C). Each dot represents an aggregate from five to seven tissue explants. (D-D″) Schematic cell contact without (left) or with (right) interstitial space in cell contact (D), and angles and tensions at the tissue surface (D′) and at the gap (D″) as in Eqns 1a,b and 2a,b. (E,F) Contact angles are $2\theta$ at the aggregate surface (E) and $2\theta_i$ at gaps (F) in normal and FN-depleted PM, from three aggregates per treatment (E) or from 19 (wild type) and 30 (FNMO) TEM sections of gastrulae (F). Averages are indicated by black bars; $P$ indicates significance levels (one-way ANOVA).

often being asymmetric (Fig. 3A), sinusoidal (Fig. 3B,C) or kinked (Fig. 3C-E). These shapes are not consistent with capillarity (Roffay et al., 2021). The most obvious deviation from a uniformly curved gap shape are the kink singularities for which a curvature $C_i$ is not even defined. Thus, $C_i$ is no longer a constraint, $\theta_i$ and $l$ can vary independently, and, of the two, only $l$ is systematically altered by FN depletion.

In view of their mechanical significance, kinks were further characterized. They occur in all gastrula tissues examined (Figs S4A-H and S5A,B) and add new tissue parameters. In two-sided gaps, most kinks are pointing outward, for $n>3$ they are oriented inward; in three-sided gaps, kinks are pointing inward and outward at equal frequency (Figs S4E-H and S5B), and inward and outward kinks can alternate on the same gap side (Fig. 3F). Apparently, kink orientation tends to fit the concave, convex or straight overall contours of gap sides. Kinks subdivide gap sides into segments whose lengths $l_{seg}$ are roughly half the gap side

lengths $l$ (Fig. S5C,D). Segments are mostly straight but occasionally concave or convex (Fig. S6A-C). The kink angle $\theta_k$ between two segments (Fig. 3F) is strikingly conserved and independent of kink orientation, $n$ or FN. Its main population is centered at 140° (Fig. 3G). The constant size of $\theta_k$ meets with the variability of segment curvature, segment length and kink orientation, which renders kinked sides adaptable. In fact, kinks do not affect contact angles $\theta_i$ (Fig. 3H,I), consistent with $\theta_i$ reflecting a mechanical balance at the gap-contact transition and kinks modifying gap surfaces independently.

Replacing a smooth by a kinked gap side at fixed $\theta_i$ and $l$ amounts to adding excess gap contour length (Fig. 3J). To see whether contours are generally longer than expected, we compared their measured lengths $S_m$ to corresponding circle segments $S_c$ calculated for the same $\theta_i$ and $l$. Ratios $S_m/S_c>1$ greatly outnumber ratios $S_m/S_c<1$ for all values of $n$ (Figs 3K,L and S7A,B); moreover, the fraction of sides with $S_m/S_c>1$ always exceeds that with kinks

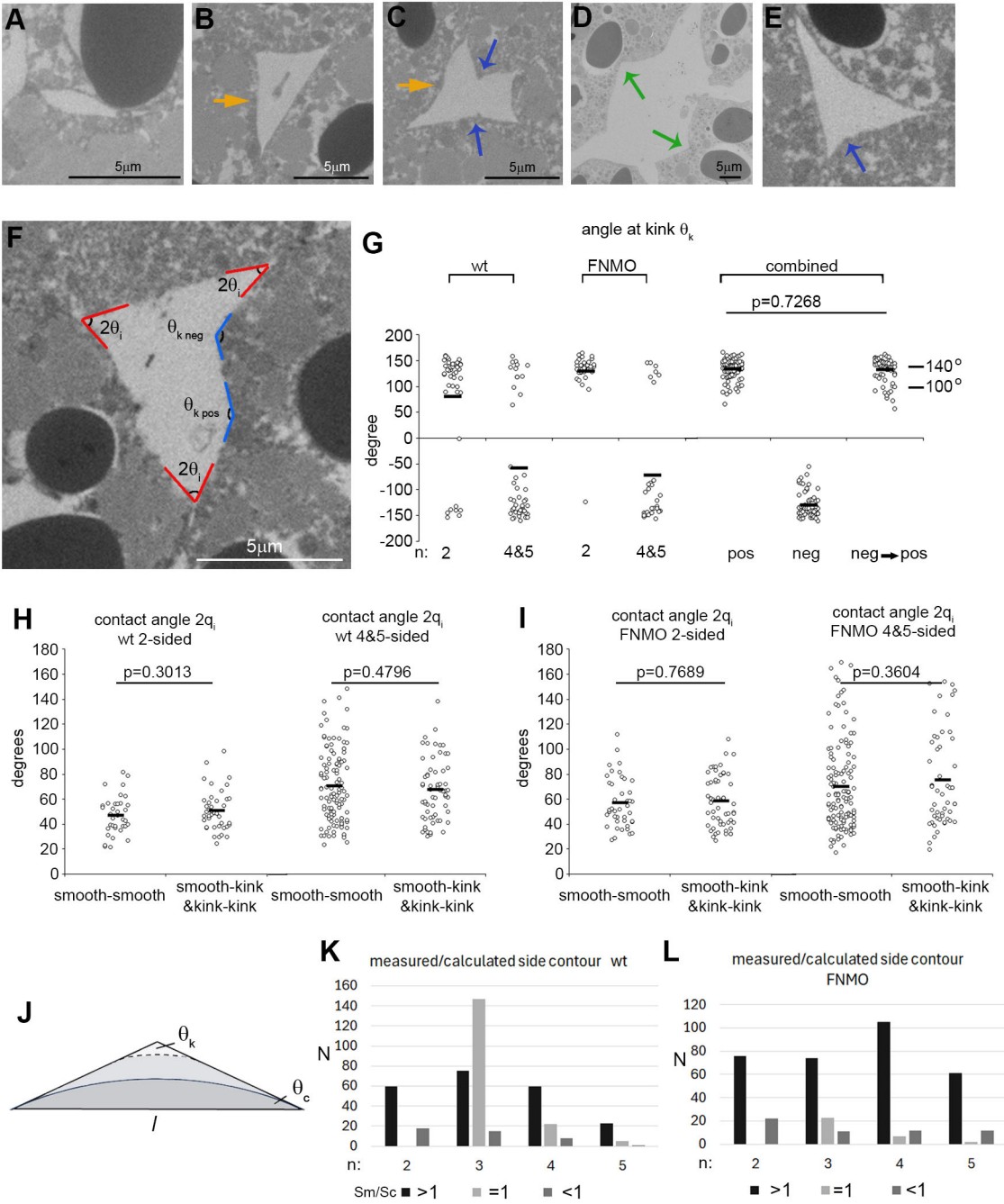

Fig. 3. Gap side shapes. (A-E) Side shapes inconsistent with capillarity. Yellow arrows indicate S-shaped contours; blue and green arrows indicate kinks. (F) A gap with contact angles $2\theta_i$ (red) and kink angles $\theta_k$ (blue) indicated. (G) $\theta_k$ from two-, and four- and five-sided gaps from normal and FN-depleted PM (21 and 28 sections). Values combined (right) with subpopulations at 140° and 100° are indicated. Averages are indicated by black bars; $P$ indicates significance (ANOVA; Mann–Whitney U-test $P$=0.5335). (H,I) Contact angles of $2\theta_i$ between smooth and kinked sides in normal PM (H) (14 and 15 TEM sections) and FN-depleted PM (I) (13 and 11 TEM sections) for two-sided, and four- and five-sided gaps. Averages are indicated by black bars; $P$ indicates significance (one-way ANOVA). (J) Schematic showing the increase of contour length at constant $l$ and angle $\theta_c$, up to kink formation at $\theta_k$. Dark grey indicates a circular contour; light grey indicates that the straight sides connected by the curved dashed line have an increased contour length for the same values of $l$ and $\theta_c$ because the contour bulges outward, and a straight-sided kink has an even longer contour. (K,L) Ratio of measured/calculated contour lengths ($S_m/S_c$) for different $n$: normal PM (K) and FN-depleted PM (L), from 14, 12, 6, 6, 13, 17, 11 and 11 TEM sections (from left to right) (see measured data points in Fig. S5).

(see Fig. S5A). This suggests that smooth gap contours are also often larger than circular minimal shapes, by bulging outward (Fig. 3J) or inward when concave. Bulging, sinusoidal and, especially, kinked shapes suggest a solid-like gap surface layer that buckles or crumples under spatial constraint.

The dynamics of surface shapes can be observed live in PM explants. While cells slowly change shape and gaps slowly grow or shrink, gap contours transition rapidly between straight, simply curved, wavy or kinked (Figs 4A-D and S8A,B). Kinks are associated with shortening gap side lengths or with contact angles at

gap corners widening or narrowing (Fig. 4C,D), but not with consistent F-actin density changes at the kink itself (Fig. S8A,B). Gaps can fuse (Fig. 4B) but can also split in two (Fig. 4C), together leaving the tissue averages of *n* and *l* presumably unaltered. Recorded at 1 frame/minute, durations of gap shapes peak at 1 min and drop rapidly within 5 min (Fig. 4E). Contacts show the same

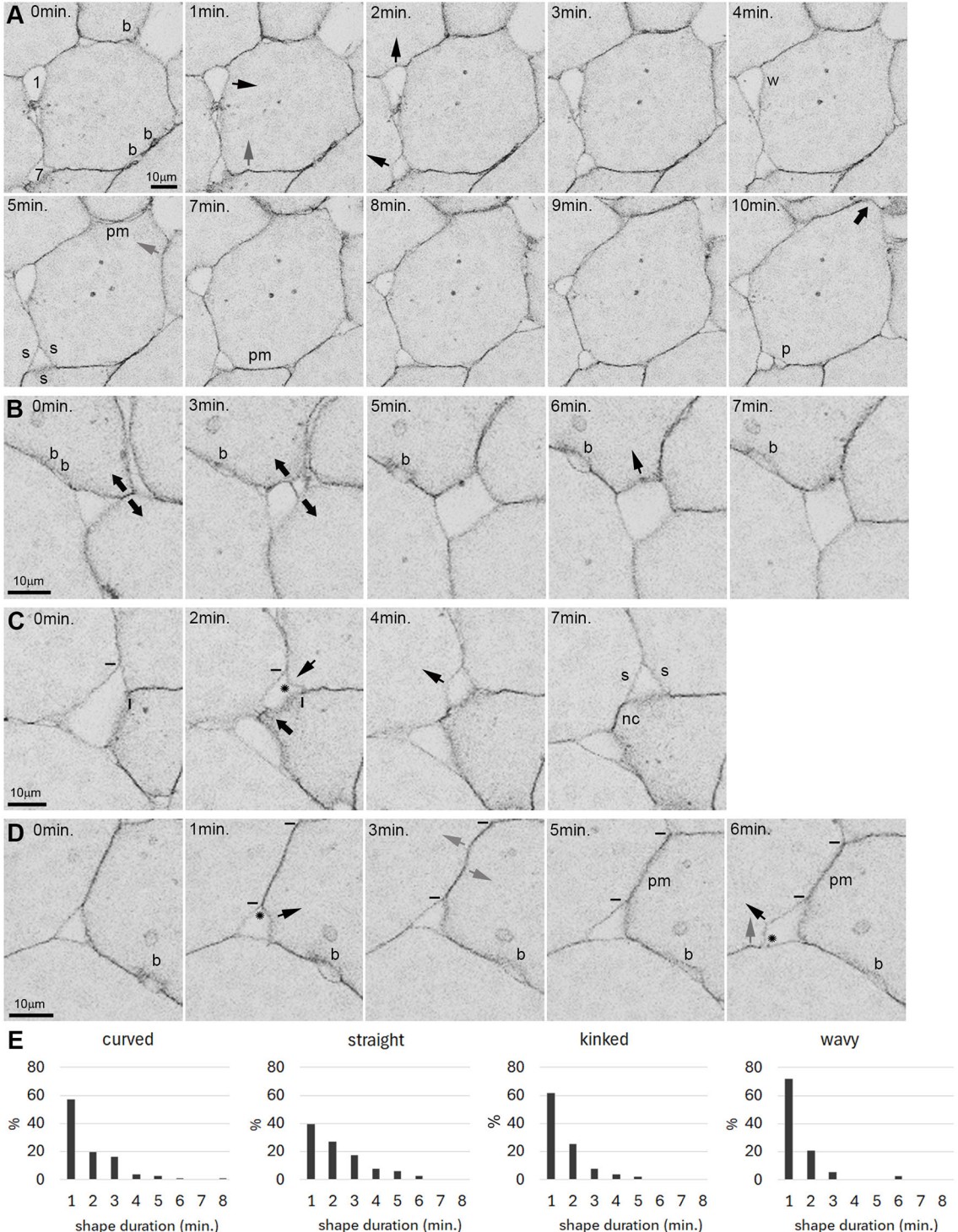

**Fig. 4. Dynamic gap side shapes.** (A-D) Sequences of time lapse frames from GFP-membrane labelled PM explants. In each panel, a selected explant region is followed through a period of time (time points are indicated in minutes), with the first of the frames shown always labelled as 0 min. Thin black arrows indicate kinks at gap surfaces; grey arrows indicate kinks at contacts; thick black arrows indicate cell movement; b, a two-sided gap; w, wavy; s, straight gap side; p, peeling apart of cells at the contact margin; pm, membrane separation (it is rare for contacts to widen over their whole extent). (A) Gap 1 grows slowly and shrinks again; gap 7 remains. (B) Contact between two three-sided gaps disappears, generating a four-sided gap. (C) New contact (nc) as cell extends across a four-sided gap, forming two three-sided gaps. (D) Kink formation near contacts; rapid contact angle changes (asterisks). Short bars in C and D indicate boundaries of contacts. (E) Durations of gap shapes: 27 gap sides followed for 17-30 min from three explants.

shape patterns as gaps (Figs 4A,D and 5A-C). Formation or loss of contact kinks is often correlated with shortening or lengthening of a surface domain (Figs 4D and 5A,B), but not with F-actin density changes at kinks (Fig. S8C,D). Instead, non-local F-actin fluctuations (Fig. S8A-D) may combine to indirectly drive shape fluctuations via surface domain size changes. Two-sided gaps appear transiently at cell contacts (Figs 4A,B,D, 5C and S8D). Contact shapes have similar lifetimes as gap shapes, but their frequency distributions are less regular and wider, with two-sided gaps having the longest lifetimes (Fig. 5D).

### Contact angle variability indicates balanced FN-depletion effects on cell-cell adhesion strength

We wondered whether FN, as an adhesion molecule, might not affect adhesion strength at all. Despite solid-like surface behavior, $\theta_i$ would still reflect the mechanical balance of tensions at gaps and thus relative adhesiveness $\alpha=1-\cos\theta_i$. To determine whether FN impacts are hidden within the exceedingly wide distribution of $\theta_i$ (see Fig. 2F), we first used the range of $n$ to disentangle the $\theta_i$ distributions of normal and FN-depleted PM. Indeed, the average $\theta_i$ suddenly increases from $n\leq3$ to $n\geq4$ in normal and in FN-depleted tissue (Fig. 6A,B), pointing at subpopulations in the overall $\theta_i$ distribution. Some of these react differently to FN. For example, FN

depletion lowers $\theta_i$ in three-sided gaps through the loss of high-$\theta_i$ subpopulations (Fig. 6A,B).

Details of the change at three-sided gaps are revealed when the data points, plotted as $\alpha$, are spread out along an additional axis provided by the contact width $w$ (Fig. 6C,D) (Luu et al., 2024). FN depletion expands the range of $w$ several-fold, but more than half of contact-gap transitions disappear, consistent with a loss of three-sided gaps (Fig. S1B,C). This is associated with a complete loss of $\alpha>0.293$ (Fig. 6C,D), i.e. of $2\theta_i>90°$ (Fig. 6A,B; red dots in Fig. 6A,C for orientation). At $60°<2\theta_i<90°$, data points are diminished in morphants (Fig. 6A,B) and the $\alpha$-$w$ plot indicates two apparently FNMO-resistant subpopulations, one with small $w$ and higher $\alpha$ (green), the other with much larger $w$ (orange) (Fig. 6B,D). The frequency of gaps with $n>3$ increases upon FN depletion (Fig. S1B,C) and, strikingly, contacts with low or high $\alpha$ are added evenly over the whole width range (Fig. 6E,F), balancing to some extent the loss of high $\alpha$ values at three-sided gaps. In summary, $\theta_i$ values form discernible subpopulations that differ in their sensitivity to FN depletion. However, depletion-induced changes in $\theta_i$ are not all biased towards reduction, and the increases and decreases leave the $\theta_i$ average unchanged overall. Having determined the magnitudes of tensions, we can now extend similar previous conclusions, tentatively based on relative adhesiveness $\alpha$

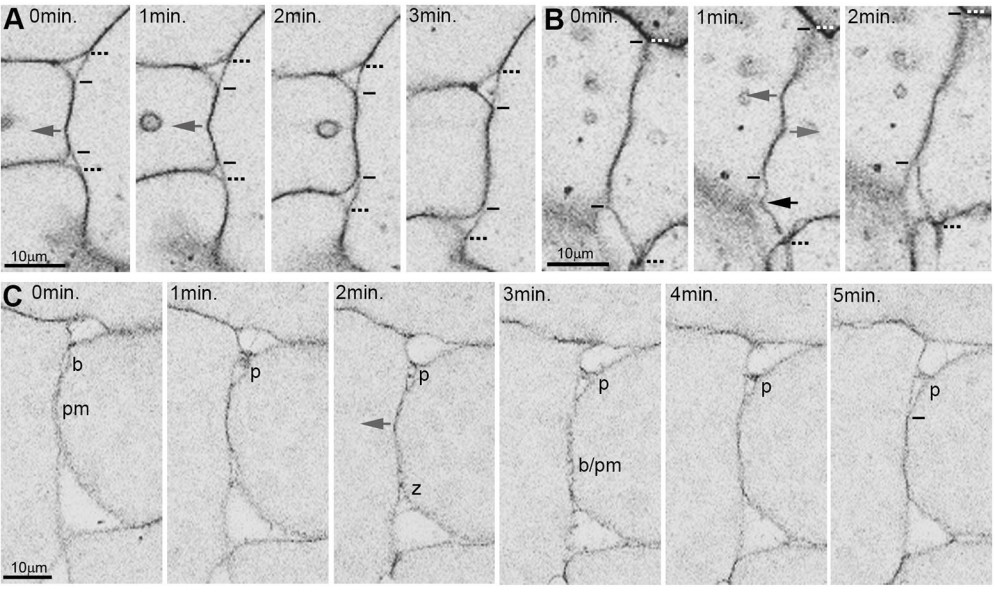

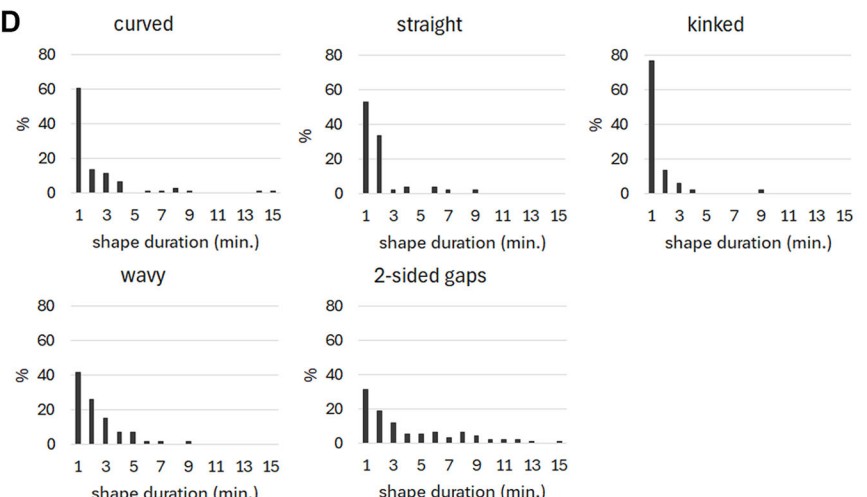

**Fig. 5. Dynamic contact shapes.** (A-C) Time lapse frames. Black arrows indicate kinks at gap surfaces; gray arrows indicate kinks at contacts; b, a two-sided gap; p, peeling apart of cells at the contact margin; pm, membrane separation (it is rare for contacts to widen over their whole extent); z, zipping up of membranes at contact margins; bars indicate boundaries of expanding or shrinking contacts; dashed bars indicate surface domains with gaps and contact. (D) Duration of contact shapes, 26 contacts followed for 17-30 min each using the same explants as in Fig. 4.

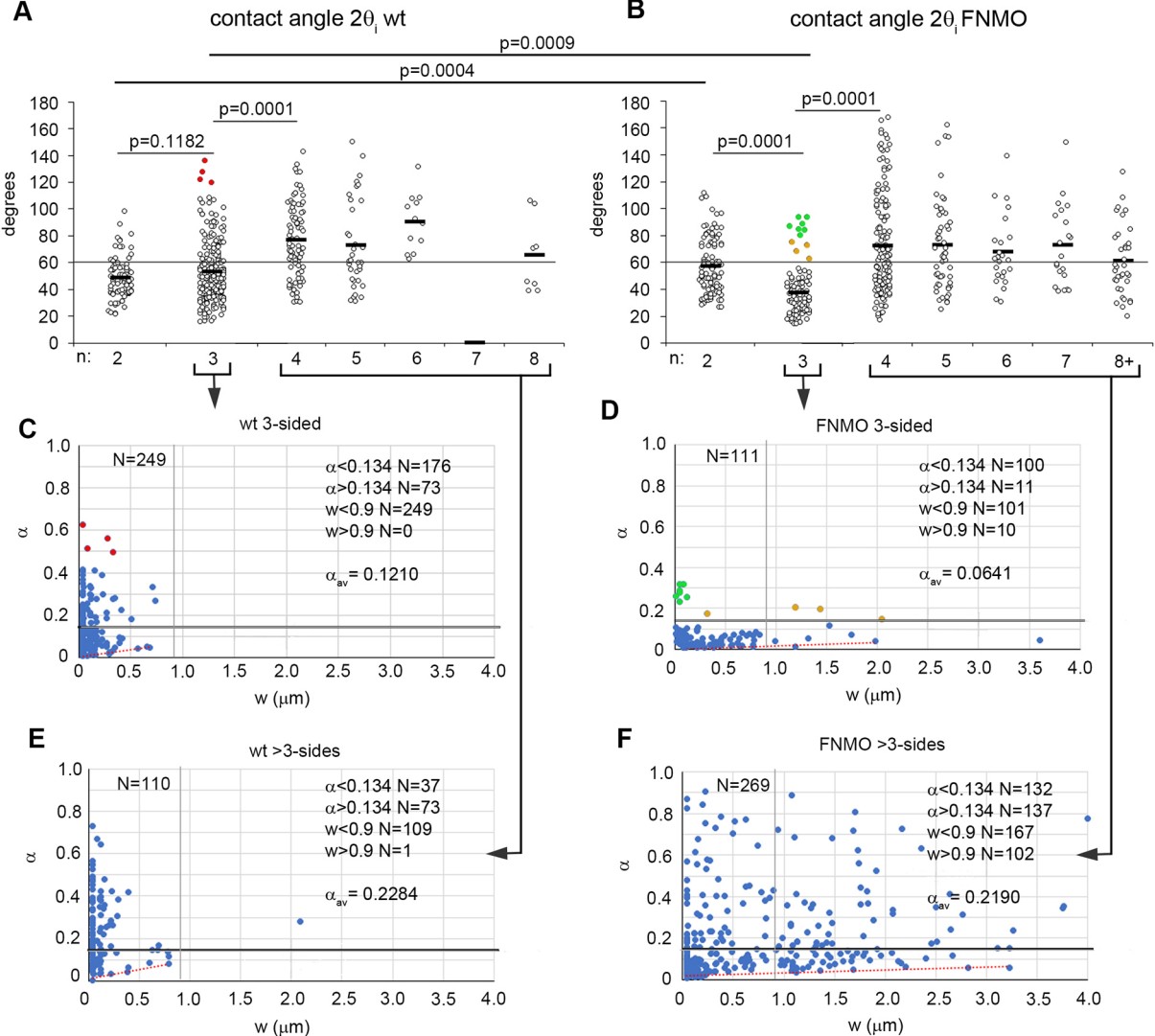

**Fig. 6. Contact angle subpopulations.** (A,B) Angles of $2\theta_i$ at the corners of *n*-sided gaps: normal PM (A) and FN-depleted PM (B). Horizontal grey reference line is at 60°. Averages are indicated by black horizontal lines; *P* indicates significance (one-way ANOVA). Data are from 24 (A) and 30 (B) TEM sections. (C-F) Relative adhesiveness ($\alpha$) versus contact width (*w*) contacts ending in gaps with *n*=3 (C,D) or *n*>3 (E,F) normal PM (C,E) or FN-depleted PM (D,F). Grey vertical reference lines indicate *w*=0.9 µm; horizontal reference lines indicate $\alpha$=0.134 (i.e. 60°); red dashed lines indicate a slight increase of lower boundary of distribution with *w*. Corresponding data points are labelled in A and C: red indicates an absence in B,D. Green and orange points in B and D represent different subpopulations in which *w* is increased (orange) or not affected (green) by FN depletion. *N*, number of measurements; *N* values for specific $\alpha$ and *w* ranges are indicated on the right; $\alpha_{av}$ indicates average of total. Data are from (C) 12, (D) 17, (E) 6; and (F) 6 TEM sections analyzed from 6 (C,E) and 3 (D,F) embryos.

(Barua et al., 2017, 2021; Barua and Winklbauer, 2022; Luu et al., 2024), to the absolute local adhesion strength $\sigma_i$ (see supplementary Materials and Methods), showing that it is indeed modulated by FN.

### Modelling biphasic elasto-capillary tissue

We propose to model gap formation as emerging naturally and together with surface stiffness from a PCM adhesion process that links gap size to PCM abundance. In the *Xenopus* gastrula, PCM layers on two converging gap surfaces can be seen to merge into a single layer in a contact. Notably, the fused layer is as high as each of the separate layers in the gap, suggesting interpenetration of PCMs at contacts (Barua et al., 2021; Luu et al., 2024). Analysis of a similar process in human blood cells *in vitro* showed that not only the height but also the density of the layers is preserved in contacts (Soler et al., 1998), implying a twofold expansion in the area of the

fused layer. Area expansion upon PCM-PCM interpenetration is a basic feature of the following model that links contact abundance *c* (fraction of cell surface in contacts) to PCM abundance *q* (fraction of cell surface covered by PCM) (Fig. 7A,B).

PCM-PCM interpenetration seems to occur by ~100 nm supramolecular PCM units (Weinbaum et al., 2007; Luu et al., 2024), whose detailed structure, shape or deformability are not known. But to convey the basic concept, PCM units can be modeled as regular 'stubs' (Fig. 7A) whose interdigitation is driven by the release of binding energy *B*/2 per surface area and per cell when stubs interact molecularly (Luu et al., 2024). We assume that stubs always remain firmly attached to the cell surface, and cell attachment or separation occur between stubs. At low initial stub density $q_0$ in a given cell surface domain, stubs will coalesce into patches to maximize adhesion, forming contacts and leaving

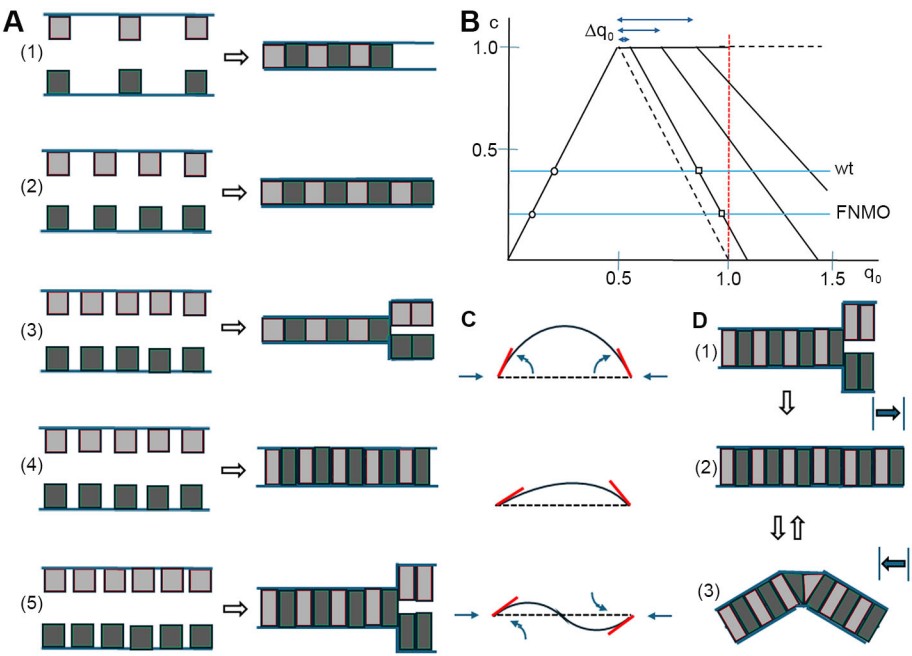

**Fig. 7. PCM adhesion.** (A) Stub interdigitation. (1) Stubs density $q_0<0.5$ leads to contact patches and stub-free surfaces. (2) At $q_0=0.5$, the whole domain is in contact. (3) Non-deforming stubs: $q_0>0.5$ leads to contact patches and excess stubs in gaps. (4) Compressible stubs: contacts form at $q_0>0.5$. (5) At higher stub densities, interdigitation ceases and compressed stubs form gaps. (B) Fraction of cell surface engaged in contacts (c) as a function of stub density ($q_0$). Initial increase until c=1 at $q_0=0.5$ (1 and 2 in A). With non-deforming stubs, c decreases for $q_0>0.5$ (dashed line) (3 in A). Compression permits interdigitation beyond $q_0=0.5$ (4 in A). After the limit of interdigitation at $q_0=0.5+\Delta q_0$, c decreases (5 in A). Blue lines indicate normal and FN-depleted PM; red dashed line indicates $q_0=1$. (C) Bending (curved arrows) or buckling (straight arrows) upon compression of stiff gap surfaces at given side lengths (dashed lines), angles (dashed lines, red tangents) and contour lengths (solid lines). (D) Stretching of a surface domain (1) can remove a gap by additional stub interdigitation (2) (white arrow). Compression of the expanded domain leads to buckling (3). Alternating stretching and compression (black arrows on the right) causes fluctuating buckling [white arrows between (2) and (3)].

remaining surfaces 'naked' and non-adhesive (Fig. 7A). Contact abundance c grows proportionally with $q_0$, and naked gaps decrease until, at $q_0=0.5$, all stubs interdigitate and the whole domain forms a contact (c=1) (Fig. 7A,B).

If non-deformable, excess stubs with no room to interdigitate at $q_0>0.5$ would accumulate and cover gap surfaces, and c would decrease (Fig. 7A,B). In an elastic PCM, lateral compression of stubs together with some area expansion and consequent out-of-plane deformation of PCM layers will permit interdigitation beyond $q_0=0.5$ (Fig. 7A,B). In compressed stubs, release of binding energy upon contact may differ, changing from $B/2$ to $B'/2$ (see supplementary Materials and Methods). With elastic deformation, $B'/2$ will induce a surface pressure $\Pi$ (Milner et al., 1989) in a PCM that is continuous in contacts and gaps. A limit for interdigitation is reached at $q_0=0.5+\Delta q_0$, when the work per area needed to deform the PCM any further equals $B'/2$. At the critical $\Pi_{crit}$, where interdigitation ceases to be supported, the PCM is modeled to have become non-adhesive by compression, analogous to the vanishing of free surface energy, or adhesion potential, in compressed monolayer films (Milner et al., 1989). When $q_0$ grows further, c must decrease, here assumed linearly, as ever more stubs are accumulating in non-adhesive patches (Fig. 7A,B).

In both normal and FN-depleted embryos, PCM is continuous in contacts and gaps (Barua et al., 2021; Luu et al., 2024). In the model, such absence of naked gaps requires that an already high $q_0$ increases further upon FN knockdown to reduce c (Fig. 7B). In fact, depletion of C-cadherin, hyaluronic acid or, indeed, FN, raises PCM deposition in gaps and contacts in the embryo (Luu et al., 2024). We conclude that gap surfaces in the PM are non-adhesive, not by being devoid of PCM but by the PCM being compressed into a non-adhesive state.

A predicted consequence of PCM compression is the transition to solid-like behavior as, for example, in surfactant monolayers (Milner et al., 1989), surface-grafted polymer brushes (Lei et al., 2015) or layers of microgel particles (Gerelli et al., 2024). Bending stiffness $\kappa$ in monolayer films increases rapidly with grafting

density q, as $\kappa\sim q^{2.2}$ to $\kappa\sim q^3$ (Milner et al., 1989; Lei et al., 2015). In the model, PCM-PCM interpenetration continues until free energy is minimized, and equilibrium gap size and shape are determined by surface tensions and elastic energies. At this point, surface pressure and surface area have increased, to a balance determined by compressive and bending stiffnesses. This causes deformation of surface shapes by buckling (in-plane compression) and bending (out-of-plane torque), as in the 'reduced-volume'-effect in lipid vesicle models where deviation from spherical shapes is induced by an increase in membrane area relative to vesicle volume, and overall energy minimization determines shapes (Lipowsky and Seifert, 1991; Ziherl and Svetina, 2007, 2008; Sackmann and Smith, 2014; Murakami et al., 2020). Respective reduced-volume shape effects appear in gastrula tissue as smooth or kinked bulging of gap surfaces. They are induced already by moderate compression of the cell surface; when isolated, spherical gastrula cells become dumbbell shaped upon forming pairs in vitro (Parent et al., 2024). Thus, elasticity must be added to surface tensions in the model. As seen in the following, this no longer allows the determination of all variables in the equations from the available data; however, importantly, all adhesion strength parameters are left unaffected by this modification.

Eqns 1a,b and 2a,b can describe adhesion in capillarity-based biphasic tissues. The PCM-PCM binding energy released per area and per cell, $B/2$, is fully translated into an adhesion tension $\Gamma/2$ per cell that drives the mutual attachment of cells, apart from any cortical tension differences (Manning et al., 2010). $\Gamma/2$ in turn equals $\sigma_i=\beta_f - \beta_c$ (Eqn 2a) (Barua et al., 2017) and

$$\frac{B}{2} = \frac{\Gamma}{2}. \tag{3}$$

With Eqn 3, Eqns 1a,b and 2a,b can be re-written as functions of $B/2$ (see supplementary Materials and Methods).

In the elasto-capillary model, not only adhesion tension $\Gamma'$ but also elastic energies per area are derived from $B'$ of the PCM-PCM interaction. $B'$ is released in contacts only where it generates

DEVELOPMENT

adhesion tension $\Gamma'$ and elastic energy at contacts, $W_c$; however, via the surface pressure $\Pi$ that spreads in the PCM, $B'$ also provides elastic energy $W_g$ in gaps and per cell (see supplementary Materials and Methods):

$$\frac{B'}{2} = \frac{\Gamma'}{2} + \frac{W_c}{2} + W_g. \qquad (4)$$

Transforming respective parameters accordingly and noting that elastic energy $W_s$ at the tissue surface may differ from $W_g$ (see supplementary Materials and Methods), Eqns 1a,b and 2a,b eventually become:

$$\sigma = (\beta' - \beta'_f) + \left(\frac{B'}{2} - (W_g + W_s)\right) \qquad (5a)$$

$$\cos\theta = \frac{\beta'_f - \left(\frac{B'}{2} - W_g\right)}{\beta' - W_s} \qquad (5b)$$

$$\sigma'_i = \frac{B'}{2} - 2W_g \quad \text{and} \qquad (6a)$$

$$\cos\theta_i = \frac{\beta'_f - \left(\frac{B'}{2} - W_g\right)}{\beta'_f - W_g}. \qquad (6b)$$

Values of $\sigma$, $\theta$ and $\theta_i$ are measured and therefore identical in both models. $\beta'$, $\beta'_f$ and $B'/2$ assume different values when elasticity is added, while local adhesion strength $\sigma'_i = \sigma_i$ (see supplementary Materials and Methods). If $B'/2$ and $W_g$ are independent, $\theta_i$ can be modulated by their relative magnitudes (Eqn 6b). High or low $B'/2$ can be compensated for by low or high $W_g$ to keep $\theta_i$ constant, i.e. changes in PCM-PCM binding energy can be neutralized by changes in elastic properties of the PCM. Hence, in FN morphants, compensation of adhesion strength effects may not only occur between different subpopulations of contacts but also between $B'/2$ and $W_g$: a weakened PCM interaction could be balanced by higher elastic energy. In addition, the increase of $\theta_i$ between $n \leq 3$ and $n \geq 4$ could reflect an increase of $W_g$ at constant $B'/2$. Notably, the values of tissue adhesion strength $\sigma$, local adhesion strength $\sigma_i$ and relative adhesiveness $\alpha$ do not change between models (see supplementary Materials and Methods). In other words, the reversible work required to create free cell surface from contacts is the same, whether it contains elastic components or not and, currently, whether FN is depleted or not.

In the model, reduced-volume effects follow directly from the PCM adhesion process, and a respective expansion of the PCM layer causes deviation of gap shapes from circular contours. Depending on the effective amount of PCM material deposited, gap surfaces are predicted to bulge to different degrees for given $\theta_i$ and $l$, with kinks as extremes (Fig. 7C top; see also Fig. 3J). Moreover, angles at the two ends of a gap side usually differ, causing asymmetrically curved shapes (Fig. 7C middle); when these angles are severely misaligned by the orientations of contacts at a gap, sinusoidal contours form (Fig. 7C bottom). Among all this variability, the kink angle $\theta_k$ is notably constant in the embryo. It is reminiscent of an angle of the same constant magnitude, $140°$, that encloses the buckled region of a generic crumpling singularity: the d-cone (Cerda and Mahadevan, 2005; Liang and Witten, 2006).

As to the dynamics of surface shapes, we note that rapid cortex fluctuations are ubiquitous in animal cells (Gorfinkiel and Blanchard, 2011; Kruse et al., 2024), and in *Xenopus* gastrula explants, cortical

F-actin density undulates at a minute timescale, over domains of a fraction of the cell perimeter (Kim and Davidson, 2011; Huebner et al., 2021; Parent et al., 2024). We propose that corresponding tension fluctuations are responsible for the rapid pace of gap and contact shape transitions. Fluctuations are asynchronous between the cells around a gap and across free and contacting surfaces, and thus generate tension balances that fluctuate in complex interference patterns. However, balances go through the same constellations of contact angles, contact orientation and contour length, yielding the same shape types listed in Fig. 7C for gaps and contacts. In addition, we propose that compression induces local delamination and buckling within contacts to form short-lived two-sided gaps that are analogous to blisters at attached thin films (Hutchinson, 1996; Yuan et al., 2019). Their elongated shape differs from the similarly dynamic, yet spherical 'inverse blebs' in mouse blastocysts produced by high interstitial pressure (Schliffka et al., 2024).

Buckling and crumpling are induced by the spatial confinement of thin elastic layers (Cerda and Mahadevan, 2003; Witten, 2007; King et al., 2012; Timounay et al., 2020), and thus the rapid shape changes of cell surfaces by the fluctuating changes of confinement due to cortex contraction and relaxation (Fig. 7D). Independent of this, the cell cortex could theoretically also contribute to surface stiffness. For example, it could stiffen locally when its growth for some time exceeded its contraction, in order to permit crumpling. However, no local cortex thickening at kinks was seen. Also, artificial cortices can buckle *in vitro*, but they do so in the absence of normal, rapid cortex turnover (Ito et al., 2015; Kusters et al., 2019; Livne et al., 2024). The time and length scales of buckling and crumpling in PM explants require surface stiffness over minutes and micrometers, but actomyosin half-life as well as tension decay times in cortices last fractions of a minute (Fritzsche et al., 2013; Fischer-Friedrich et al., 2016; Saha et al., 2016) and bending stiffness is negligible above tens of nanometers (Murrell and Gardel, 2012; Cartagena-Rivera et al., 2016). Rapid cortex turnover (Clement et al., 2017; Lenne et al., 2021) is consistent with significantly faster dissipation in the cortex than in the PCM, and we propose that the cortex determines the timing of surface shape transitions but does not contribute substantially to surface stiffness.

## DISCUSSION

Our analysis of adhesion strength, interstitial space geometry and cell surface dynamics in the PM led to an elasto-capillary model of PCM-mediated adhesion that links PCM abundance to cell contact abundance in a biphasic tissue. It distinguishes theoretically between two modes of interstitial gap formation. At low PCM abundance, part of the cell surface would remain PCM free and, in principle, non-adhesive, forming gaps. In the high-abundance mode, adhesion entails compression of the PCM layer such that part of it becomes non-adhesive and encloses gaps whose size depends on the amount of PCM material, not on adhesion strength. This mechanism relates the augmented PCM deposition upon FN knockdown (Luu et al., 2024) to increased interstitial space; at the same time, it predicts stiffening of the cell surface. The presence of PCM in gaps, solid-like behavior of gap surfaces and contacts, and overproduction of PCM upon FN depletion strongly argue for this second mode of gap formation in the gastrula.

The PCM is molecularly and structurally inhomogeneous, and, in principle, gaps could also form if a specific fraction of the PCM were non-adhesive. This would lead to a slower increase of $c$ with $q_0$ (Fig. S9), while gaps could be covered with PCM. An increase of the non-adhesive PCM fraction in FN morphants would further lower $c$ (Fig. S9). The FN function in PCM formation is indeed

DEVELOPMENT

translated indirectly and non-uniformly into adhesion strength, increasing or decreasing it depending on contact type. We thus cannot exclude the possibility that some of the contact types become non-functional in morphants, increasing gap formation. However, if taken not as a complementary but as an alternative model, excess PCM accumulation in adhesion molecule-depleted tissue would remain an isolated fact not linked to increased gap formation. In addition, surface stiffening by compression would not necessarily ensue, and stiffness would require other explanations. An altogether different mechanism of gap formation would be hydraulic fracturing, as in mouse blastocysts, where PCM abundance does not determine gap size but the amount of interstitial fluid does (Dumortier et al., 2019). However, blastocyst gaps are concave due to high hydraulic pressure, but PM gap contours are mostly convex for $n>2$, and kinked or S-shaped. Moreover, blastocyst gaps are unstable and eventually fuse into a single lumen. Altogether, we propose that PM gaps are due to adhesion failing between opposed cell surfaces, not due to overall low adhesion strength but to an induced, spatially restricted loss of adhesion.

The independence of cell-packing density and adhesion strength has profound consequences for tissue structure, allowing the simultaneous optimization of the interstitial space and tissue cohesion. We trust that the concepts developed here will be useful in further investigations of this feature. They should also guide the study of PCM-mediated cell adhesion, a tissue cohesion mechanism that is understudied but probably widespread among metazoans (see Winklbauer, 2019, for a review). The occurrence of kinked or S-shaped contacts in diverse tissues supports this notion (Roffay et al., 2021). It may easily be substantiated by demonstrating large contact widths that ae incompatible with, for example, cadherin adhesion, and by identifying bona fide PCM components in contacts, e.g. the collagen IV that mediates adipocyte-adipocyte adhesion in *Drosophila* (Dai et al., 2017). Buckling and crumpling of interstitial surfaces should stimulate the study of PCM-generated stiffness and the elastic properties of the PCM. Together, these approaches will help to clarify the roles of the PCM both in contact and in gap formation in animal tissues.

## MATERIALS AND METHODS
### Embryo manipulations
Adult *Xenopus laevis* DAUDIN were maintained in accordance with University of Toronto Animal Use Protocol (20011765). Eggs were fertilized *in vitro*, de-jellied using 2% cysteine in 1/10 Modified Barth's Solution [MBS; 88 nM NaCl, 1 mM KCl, 2.4 mM NaHCO$_3$, 0.82 mM MgSO$_4$, 0.33 mM Ca(NO$_3$)$_2$, 0.41 mM CaCl$_2$, 10 mM Hepes (+NaOH), 1% streptomycin, 1% penicillin (pH 7.4; Sigma-Aldrich Canada)] and kept in 1/10 MBS until the required stages. Morpholino antisense oligonucleotides (Gene Tools) (Table S1) were injected at the two-cell stage in 4% ficoll solution using a Nanoinject II (Drummond Scientific Company) and embryos were incubated in 1/10 Modified Barth's Solution at 15°C until stage 11.

### Immunofluorescence staining with FN antibody
For whole-mount staining, BCR or PM was excised from stage 11 embryos fixed in 4% paraformaldehyde in MBS/BSA. Rabbit antiserum against *Xenopus laevis* plasma FN (Winklbauer, 1998) was used at 1:1000 dilution in PBS with 1% BSA, and detected with Cy3-goat-anti-rabbit IgG (1:200 dilution) (Jackson ImmunoResearch Laboratories) (Nagel and Winklbauer, 2018) in a Leica If-SP8 confocal microscope with a 40× oil immersion objective and Leica Application Suite X software.

### Tissue surface tension measurement
The procedure is outlined in Fig. S2. Five to seven PM explants excised in MBS from stage 10.5 embryos were fused into an aggregate for 1 h. Aggregates were allowed to assume a drop shape on plasticine for 2 h

and fixed with 4% formaldehyde before imaging. Tissue-surface tension was determined using Axisymmetric Drop Shape Analysis (ADSA; Del Río and Neumann, 1997; Luu et al., 2011). The drop shape of an aggregate represents an equilibrium between tissue surface tension σ and gravity, which round up and flatten the tissue, respectively. ADSA generates theoretical drop shapes for different surface tensions using the Laplace equation and finds the best fit to an aggregate outline. Afterwards, aggregates were cut in half, laid flat on a glass-bottomed dish and photographed on an inverted Zeiss IM35 microscope for measuring contact angles using Axiovision.

### Transmission electron microscopy
For transmission electron microscopy (TEM) images, 4% paraformaldehyde and 2.5% glutaraldehyde in 0.05 M cacodylate buffer at pH 7.0 were used to fix stage 11 gastrulae. Bisected samples were rinsed in 0.1 M cacodylate and further fix in 0.1 M cacodylate containing osmium tetroxide (1%). To visualize the glycocalyx, 1% lanthanum nitrate (Sigma-Aldrich Canada) was added to the fixatives. Samples were washed with 0.1 M cacodylate and dehydrated in a series of graded ethanol solutions before embedding in Spurr's resin. Semi-thin and ultrathin sections were obtained using a Lecia EM UC6 microtome. Sections were stained with 3% uranyl acetate in methanol for 1 h followed by 10 min in Reynold's lead citrate. Images were taken with a Hitachi HT7700 microscope.

### Analysis of TEM images
Cell contacts were analyzed in the same TEM images of normal and morphant stage 11 gastrulae, as previously described (Barua et al., 2021). Stretches of cell perimeter were treated as contacts when contours of adjacent cells followed each other; abrupt divergence indicated the end of a contact at a non-adhesive gap. Cell contact angles were measured as $2\theta_i$ at the transitions of contacts to interstitial gaps (AxioVision v4.8.2 and v4.9.1 image analysis software). Contact width was measured as separation distances between membranes, binned in 50 nm steps. Gap segment curvatures $C_{seg}$ were calculated as the curvatures of circular arcs between two ends of a segment (Barua et al., 2017). Arc width, W, was the distance between the segment ends, with height H perpendicular at the center of the width line and ending at the cell surface. $C_{seg}$ was then estimated using the intersecting chord theorem $1/C=(H/2)+(W^2/8H)$ (Barua et al., 2017). All identifiable contacts, angles or curvatures in a TEM image were measured; for the number of TEM images per treatment, see the figure legends. Random tilting and positioning of sectioning planes relative to cell equators, which is unavoidable in tissue sections of randomly packed tissues, introduce systematic errors. For length measurements (gap side length $l$, contact widths $w$) averages are overestimated by 1.25- to 1.5-fold (Barua et al., 2017; 2021); for contact angles, averages approximately remain the same but distributions are broadened by about 1.3-fold (Luu et al., 2024). Of note, in the present work arguments are based on relative changes in length, not on absolute values. Variances between treatments were statistically analyzed using one-way ANOVA. Data visualization and statistical analyses were performed using Graphpad Prism 7 2017 v7.0.3. Vector graphics and figures were assembled using Inkscape v0.92 and v1.0.

### Live imaging of PM explants
For membrane labelling, embryos were injected at the four-cell stage with mRNA transcribed *in vitro* from plasmid CS2+mbGFP (a gift from R. Harland, University of California, Berkeley, CA, USA). At gastrula stage 11, embryos were transferred to MBS and the vitelline membrane was removed with forceps. PM explants were excised from embryos at room temperature under a MZ16F Leica stereomicroscope. Glass-bottomed tissue culture dishes were coated with 1 mg/ml of bovine serum albumin (BSA) (Sigma-Aldrich Canada) for 20 min to prevent attachment of explants placed in the dishes in MBS and secured under a strip of coverslip. Time-lapse recordings were taken on a Leica TCS SP8 laser scanning confocal microscope with a 40× oil immersion objective. GFP tagged LifeAct in pCS2+ vector (a gift from C.-P. Heisenberg, Institute of Science and Technology Austria, Klosterneuburg, Austria) and membrane-bound-RFP in pCS2+ vector (a gift from N. Kinoshita, National Institute for Basic Biology, Okazaki, Japan) were transcribed *in vitro* and injected at 200 pg per blastomere each to visualize F-actin and cell membrane, respectively.

## Acknowledgements
We thank the CSB Imaging Facility for contributions to this work, and Tony Harris and Serge Parent for critical reading of the manuscript and many helpful suggestions.

## Competing interests
The authors declare no competing or financial interests.

## Author contributions
Conceptualization: R.W.; Formal analysis: R.W., O.L.; Funding acquisition: R.W.; Investigation: R.W., O.L., D.B., M.N., Y.H.; Project administration: R.W.; Supervision: R.W.; Visualization: O.L., M.N.; Writing – original draft: R.W.; Writing – review & editing: R.W.

## Funding
Funding was provided to R.W. by the Canadian Institutes of Health Research (PJT-15614) and by the Natural Sciences and Engineering Research Council of Canada (RGPIN-2017-06667). Open Access funding provided by the University of Toronto. Deposited in PMC for immediate release.

## Data and resource availability
All relevant data and details of resources can be found within the article and its supplementary information.

## Peer review history
The peer review history is available online at https://journals.biologists.com/dev/lookup/doi/10.1242/dev.204663.reviewer-comments.pdf

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
