## [Peer Review File · Development (Cambridge, England)]

Adhesion strength, cell packing density and cell surface buckling in pericellular matrix-mediated tissue cohesion

Rudolf Winklbauer, Olivia Luu, Debanjan Barua, Martina Nagel and Yunyun Huang
DOI: 10.1242/dev.204663

Editor: Paul Francois

Review timeline

Original submission:	17 January 2025
Editorial decision:	8 April 2025
First revision received:	4 June 2025
Editorial decision:	29 June 2025
Second revision received:	20 July 2025
Accepted:	28 July 2025

Original submission

First decision letter

MS ID#: dev.204663

MS TITLE: Adhesion strength, cell packing density, and cell surface buckling in pericellular matrix-mediated tissue cohesion

AUTHORS: Rudolf Winklbauer; Olivia Luu; Debanjan Barua; Martina Nagel; Yunyun Huang

Dear Dr Winklbauer,

I have now received all the referees' reports on the above manuscript, and have reached a decision. The referees' comments are appended below, or you can access them online: please go to:

As you will see, the referees express considerable interest in your work, but have some significant criticisms and recommend a substantial revision of your manuscript before we can consider publication. If you are able to revise the manuscript along the lines suggested, which may involve further experiments, I will be happy receive a revised version of the manuscript. Your revised paper will be re-reviewed by one or more of the original referees, and acceptance of your manuscript will depend on your addressing satisfactorily the reviewers' major concerns. Please also note that Development will normally permit only one round of major revision. If it would be helpful, you are welcome to contact us to discuss your revision in greater detail. Please send us a point-by-point response indicating your plans for addressing the referees' comments, and we will look over this and provide further guidance.

Please attend to all of the reviewers' comments and ensure that you clearly highlight all changes made in the revised manuscript. Please avoid using 'Tracked changes' in Word files as these are lost in PDF conversion. I should be grateful if you would also provide a point-by-point response detailing how you have dealt with the points raised by the reviewers in the 'Response to Reviewers' box. If you do not agree with any of their criticisms or suggestions please explain clearly why this is so.

Reviewer 1

SUMMARY OF THE ADVANCE MADE IN THIS PAPER AND ITS POTENTIAL SIGNIFICANCE TO THE FIELD

This article examines the intimacy of the cell-cell contacts in relation to adhesion and ECM in involuting pre-chordal mesoderm in vivo and in cell aggregates and analyses and models interstitial gaps and contacts to understand the interactions between cell adhesion, surface tension and pericellular matrix. The authors find that FN depletion (unlike Cad depletion) surprisingly does not change cell-cell adhesion strength, as determined by contact angles - as they reported in a previous paper, although with additional measures here - but does change gaps between cells. They then report a number of aspects of gap geometry in relation to the numbers of surrounding cells, including the occurrence of zigzagging and sinuous cell surface contours surrounding some gaps. They show that this is incompatible with an earlier simple model of gap formation and introduce a new model that includes elasticity/stiffness of pericellular matrix structures that better explains their findings. The particularly introduce the idea that pericellular crowding can reduce cell-cell adhesion and so increasing matrix above a certain surface density can create gaps.

Overall, this paper is interesting, both for the data and the modelling. It could be more succinct and there are several parts in text, figures and legends that could and should be clarified. It advances the field by focusing on cell biology in whole tissue and providing a conceptual framework for thinking about the details of dynamic cell cell interfaces.

SUGGESTIONS TO AUTHORS

The authors should consider the inhomogeneity of not only the PCM but also the cortical cytoskeleton in explaining, for example, the kinked contours. (Perhaps they see evidence of this in their TEMs?)

For the middle part of the article where modelling is first introduced it should be made clearer which experiments/analyses are of tissue in vivo and which are in ex vivo aggregates.

Fig. 1C,D: Show the drop shape analysis for the Cad depletion. (Reference should be to 2009 paper not 2014?)

Fig. 2J-J" Labelling is too small even in J' and J"

p.6, last line "attenuated downregulation" - what does this mean? Maybe just "downregulation"?

p.7 One would not expect contact angles of interior voids to correspond to polygon interior angles except under some very extreme regimes, but if one is going to make this comparison, then why not plot $\theta-f$ on the graphs in D and E? also, since $\theta-f$ for a two-vertex polygon (i.e. a line) is zero, then this must be in a different category.

Fig.3 Legend should explain the meaning of the coloured data points (panels D, E and G) and red dotted lines (panels F-I). Number of biologically independent samples (i.e. different embryos and images) should be reported in the legend, as it says they are in the Methods.

p.7-8 The main point seems to be that there is no correlation between the gap width and the contact angles. (A way of showing this would be just to plot the change in variance of gap width upon FN depletion, which contrasts with the lack of change in the contact angles, although perhaps the lack of correlation in the WT condition is significant). Perhaps this lack of correlation should be stated more explicitly as the main conclusion here.

p.8, penultimate paragraph "l is proportionally higher": Text is ambiguous. The gap lengths are just "higher" than WT for all values of n, but if this means "proportionally higher for $n \geq 3$ versus $n = 2$ or 3" then this should be spelled out. Also, in this paragraph, "non-proportionally" or, better, "non-linearly" would be clearer than "disproportional" since the latter implies an anomaly whereas the increase of A approximately with l^2 is entirely expected.

Fig. 4 Legend should make it clearer which of the panels is experimental and which is based on the model.

p.9 last paragraph: The authors should use Fig. S2 to explain more clearly the difference between the circle-segment contour and the capillary model contour derivations and how it relates to the way gap height is defined and measured.

p.9, last paragraph: Fig. 5I purports to show that "C-eff...can vary despite constant theta and l", but the figure shows one curve filled with dark grey that corresponds with the description of C-eff and another in light grey that does not, since it is not a circle segment (only the top is curved). It is not clear how C-eff is calculated or how it varies (the text says that it does). Possibly two contour shapes and their respective C-eff measures should be shown side-by-side.

p.10-11 and panels 5J, K: This is bordering on trivial, since of course continuous straighter contours are shorter than those that zigzag or are wavy.

p.11, last paragraph and panels 7E-G: This is extremely confusingly written and the axis labels are insufficient. Why, for example, are (average?) duration and half-life plotted separately since presumably they are trivially dependent variables?

p.12 top: PCM abundance q should be more precisely defined as cell surface fractional PCM occupancy.

Fig. 8B legend should (again) define c

p. 12 last paragraph: "As $B/2$ deforms stubs, it may itself be altered (Suppl. Text), and $B'/2$ induces a surface pressure Π (Milner et al. 1989) in a PCM which is continuous in contacts and gaps.". " $B/2$ " and " $B'/2$ " should be defined here.

p.14: This text might be a bit clearer if the qualitative conclusions were stated first and the physical calculations followed (introduced with phrases such as "this is based on the following:").

Reviewer 2

Following up on previous work, Winklbauer et al., explore the influence of the pericellular matrix (PCM) on cell adhesion and tissue packing by performing a detailed analysis of the contacts and interstitial gaps found between cells within *Xenopus* embryonic tissue. Using a fibronectin morphant tissue with reduced levels fibronectin but where PCM deposition is enhanced, they observe that the size of gaps between cells are increased. They measure that cell-cell contact size is decreased in the morphant, but counterintuitively, adhesion strength, deduced from tissue surface tension and contact angle measurements of explants, would not be affected. They go on to show that the existing capillary models used to describe adhesion are not sufficient to explain the shape and size of interstitial gaps. They propose a new elasto-capillary model that considers the elasticity of the PCM to better capture the properties of cell-cell adhesions and the interstitial gap network. While the concepts proposed here are interesting, the current presentation of both the text and the figures are hard to follow, especially for readers unfamiliar with the literature, and there should be significant restructuring of the manuscript to make the work accessible.

The manuscript is rather long and the figures rather large. As far as we understand, the main conclusion is that elasticity must be taken into consideration to explain the shape of gaps within the tissue. Before reaching this conclusion based on the data shown in Fig5, the authors describe at length a capillary model before explaining that this description is not appropriate. Actually, a large part of the data and analysis shown in Fig 1 to 4 becomes irrelevant to the conclusion since they rely on the assumption that the tissue cannot be simply considered as liquid-like. Concretely, many considerations (e.g. tensions at gaps deduced from angles, of tissue surface tension, of mean angle comparisons that turn out to not be normally distributed and cannot be compared as such) end being irrelevant for the conclusions. This lack of hierarchy in the presentation of data makes it very difficult for the reader to follow the authors train of thought.

Similarly, some of the figures are rather large (more than 10 panels) and often contain redundant information (e.g. average value of angle with 2 decimals given below a graph that shows individual measurements as well as the average) or information that may not be essential enough to be shown in a main figure (what is the benefit of showing the brackets, or coloured dots in fig 3D?). At the same time, some data is shown incompletely (e.g. fig 1 gives mean and p values but not the

dispersion of the data, fig 2 no image of control vs morphant aggregates is shown or the raw image used to compute surface tension) or incorrectly shown (e.g. fig 5j-k shows the ratio of surface as a function of side length when the relevant information is the distribution given as numbers pasted on the graph, control and morphant measurements should be displayed side by side rather than in separated graphs for readers to be able to compare them). Again, the lack of proper hierarchy in the data that is displayed makes it difficult for the reader to follow the relevant information. We think that the manuscript would very much benefit from a clearer organization before we could comment on the other smaller issues such as: measurements (mostly on fixed tissue when the gaps are clearly dynamic, often at unknown depth of the contact between cell when the equatorial plane should probably be considered, tension measurements should be given in SI units i.e. N/m and not J/m²), statistical (not all data is normally distributed) or conceptual (seeing how dynamic the gap shape is, how is the cytoskeleton considered here? The cytoskeleton would be equally likely to the PCM to provide elastic contribution to shape the kinks at gaps) considerations.

First revision

Author response to reviewers' comments

We sincerely thank the reviewers for their patience with our manuscript. We were obviously confused by results that required us to abandon our previous capillarity-based view of tissue cohesion. As requested, we completely restructured the Results section of the paper including the figures. We also added new data on cell cortex behavior at kinks.

Reviewer 1: SUMMARY OF THE ADVANCE MADE IN THIS PAPER AND ITS POTENTIAL SIGNIFICANCE TO THE FIELD

This article examines the intimacy of the cell-cell contacts in relation to adhesion and ECM in involuting pre-chordal mesoderm in vivo and in cell aggregates and analyses and models interstitial gaps and contacts to understand the interactions between cell adhesion, surface tension and pericellular matrix. The authors find that FN depletion (unlike Cad depletion) surprisingly does not change cell-cell adhesion strength, as determined by contact angles - as they reported in a previous paper, although with additional measures here - but does change gaps between cells. They then report a number of aspects of gap geometry in relation to the numbers of surrounding cells, including the occurrence of zigzagging and sinuous cell surface contours surrounding some gaps. They show that this is incompatible with an earlier simple model of gap formation and introduce a new model that includes elasticity/stiffness of pericellular matrix structures that better explains their findings. The particularly introduce the idea that pericellular crowding can reduce cell-cell adhesion and so increasing matrix above a certain surface density can create gaps.

Overall, this paper is interesting, both for the data and the modelling. It could be more succinct and there are several parts in text, figures and legends that could and should be clarified. It advances the field by focusing on cell biology in whole tissue and providing a conceptual framework for thinking about the details of dynamic cell cell interfaces.

The paper, in particular the Results section was re-written to stress the overall argument and to clarify various points. Data not related to the main line of argumentation were removed, the number of figures and figure panels was reduced, and the remaining data were ordered more transparently.

SUGGESTIONS TO AUTHORS

The authors should consider the inhomogeneity of not only the PCM but also the cortical cytoskeleton in explaining, for example, the kinked contours. (Perhaps they see evidence of this in their TEMs?)

Inhomogeneity of the cortical cytoskeleton in this system has been described by others and us (see

references in the manuscript) as spatio-temporal density and activity fluctuations and we used this in our model to explain the fluctuations of cell surface shapes including kinks. We assumed that complex interference patterns from independent fluctuations in different cells, contacts, and contact sections would drive fluctuations of resultant tensions, which in turn would affect the shapes of cell surfaces characterized by a contractile cortex and a stiff PCM. We add now new data (Fig.S6) suggesting that, in agreement with this view, cortical F-actin is not usually acting locally at sites of kinking to cause these curvature singularities, e.g. by specific, local changes of cortex density or the F-actin pattern.

For the middle part of the article where modelling is first introduced it should be made clearer which experiments/analyses are of tissue in vivo and which are in ex vivo aggregates.

Experiments/analyses are either in fixed whole embryos (not really in vivo) or in living explants of prechordal mesoderm. We make this clear now where appropriate. We also indicate what is experimental observation and what is postulated in or derived from the model.

Fig. 1C,D: Show the drop shape analysis for the Cad depletion.

Figs.1A-D merely illustrated technical details already published elsewhere and were deleted in the course of de-cluttering the figures. Fig.1C,D show just examples of a step in the ADSA procedure. For Cad depletion this was shown e.g. in Ninomiya et al. 2012, J. Cell Sci. 125, 1877-1883, Fig.2D-G.

(Reference should be to 2009 paper not 2014?)

Reference is to the 2014 paper where σ and contact angles were measured, and tensions were calculated for several gastrula tissues.

Fig. 2J-J" Labelling is too small even in J' and J"

The figure was restructured and font size increased.

p.6, last line "attenuated downregulation" - what does this mean? Maybe just "downregulation"?

Was replaced by "a lesser downregulation of beta".

p.7 One would not expect contact angles of interior voids to correspond to polygon interior angles except under some very extreme regimes, but if one is going to make this comparison, then why not plot $\theta-f$ on the graphs in D and E? also, since $\theta-f$ for a two-vertex polygon (i.e. a line) is zero, then this must be in a different category.

We think that the contact angles, in so far as they indicate relative adhesion strength, do indeed "attempt" to stay relatively constant over n , as seen in former Fig.3D,E, now Fig.6A,B. The polygon depiction has now been moved to the supplement though (Fig.S1A), together with former Fig.3BC (now Fig.S1B,C). And indeed, we had not treated the 2-sided gaps as a separate category here; we corrected this (see now Fig.S1B,C).

Fig.3 Legend should explain the meaning of the coloured data points (panels D, E and G) and red dotted lines (panels F-I). Number of biologically independent samples (i.e. different embryos and images) should be reported in the legend, as it says they are in the Methods.

The whole respective paragraph has been rewritten for clarity, and the meaning of the coloured data points and the dotted lines is explained now in the main text and the figure legend. Numbers are indicated as requested.

p.7-8 The main point seems to be that there is no correlation between the gap width and the contact angles. (A way of showing this would be just to plot the change in variance of gap width upon FN depletion, which contrasts with the lack of change in the contact angles, although perhaps the lack of correlation in the WT condition is significant). Perhaps this lack of correlation should be stated more explicitly as the main conclusion here.

The intention here was to further identify subpopulations (which might react differently to FN depletion) by adding w as additional “dimension” (the initial one being n). For example, at $n = 3$, the orange and green dots represent different subpopulations as seen from the fact that orange, but not green data points respond to FN depletion by increasing w . Or the different responses to FN depletion at $n > 3$. All this is to show that the adhesion molecule FN does indeed affect adhesion, not uniformly though but such that on average, effects approximately cancel out. We stress this point now more clearly.

p.8, penultimate paragraph “l is proportionally higher”: Text is ambiguous. The gap lengths are just “higher” than WT for all values of n , but if this means “proportionally higher for $n > 3$ versus $n = 2$ or 3 ” then this should be spelled out.

We replaced this by “in FN morphants l is increased at each n ”.

Also, in this paragraph, “non-proportionally” or, better, “non-linearly” would be clearer than “disproportional” since the latter implies an anomaly whereas the increase of A approximately with l^2 is entirely expected.

We removed the gap area A -sidedness n plots as this was not contributing to the main argument, and was indeed “entirely expected” anyway.

Fig. 4 Legend should make it clearer which of the panels is experimental and which is based on the model.

Former Fig.4 was dissolved. Fig.4A,B became Fig.1K,L; old Fig.4C,D was deleted as not contributing essentially; E-H became obsolete with the removal of C_{eff} and was also removed; and old Fig.4I-M was moved to Fig.3A-E. It should now be clear which panels show experimental results.

p.9 last paragraph: The authors should use Fig. S2 to explain more clearly the difference between the circle-segment contour and the capillary model contour derivations and how it relates to the way gap height is defined and measured.
and also

p.9, last paragraph: Fig. 5l purports to show that “ C_{eff} ...can vary despite constant θ and l ”, but the figure shows one curve filled with dark grey that corresponds with the description of C_{eff} and another in light grey that does not, since it is not a circle segment (only the top is curved). It is not clear how C_{eff} is calculated or how it varies (the text says that it does). Possibly two contour shapes and their respective C_{eff} measures should be shown side-by-side.

and also

p.10-11 and panels 5J, K: This is bordering on trivial, since of course continuous straighter contours are shorter than those that zigzag or are wavy.

We wanted to show (now Fig.3J) that simply curved surfaces, not only kinked or wavy ones, are also longer than expected (bulging outward compared to circular outlines). Former Fig.5J,K has been moved to the supplement (Fig.S5A,B) to show the original data points, but the relevant summary data are in the new Fig.3K,L. The C_{eff} data, remnants of the old capillarity model, have been removed as too peripheral to the main argument.

p.11, last paragraph and panels 7E-G: This is extremely confusingly written and the axis labels are insufficient. Why, for example, are (average?) duration and half-life plotted separately since presumably they are trivially dependent variables?

These panels did indeed not contribute much to the understanding of shape fluctuations, and were removed.

p.12 top: PCM abundance q should be more precisely defined as cell surface fractional PCM occupancy.

Done.

Fig. 8B legend should (again) define c

Done

p. 12 last paragraph: "As B/2 deforms stubs, it may itself be altered (Suppl. Text), and B'/2 induces a surface pressure Π (Milner et al. 1989) in a PCM which is continuous in contacts and gaps.". "B/2" and "B'/2" should be defined here.

B'/2 is shortly and very generally defined in the main text but fully explained in the supplementary text, for not to burden the main text with details that must be hypothetical at this time.

p.14: This text might be a bit clearer if the qualitative conclusions were stated first and the physical calculations followed (introduced with phrases such as "this is based on the following:").

We now stated the main qualitative conclusion at the end of the preceding paragraph.

Reviewer 2: Following up on previous work, Winklbauer et al., explore the influence of the pericellular matrix (PCM) on cell adhesion and tissue packing by performing a detailed analysis of the contacts and interstitial gaps found between cells within Xenopus embryonic tissue. Using a fibronectin morphant tissue with reduced levels fibronectin but where PCM deposition is enhanced, they observe that the size of gaps between cells are increased. They measure that cell-cell contact size is decreased in the morphant, but counterintuitively, adhesion strength, deduced from tissue surface tension and contact angle measurements of explants, would not be affected. They go on to show that the existing capillary models used to describe adhesion are not sufficient to explain the shape and size of interstitial gaps. They propose a new elasto-capillary model that considers the elasticity of the PCM to better capture the properties of cell-cell adhesions and the interstitial gap network. While the concepts proposed here are interesting, the current presentation of both the text and the figures are hard to follow, especially for readers unfamiliar with the literature, and there should be significant restructuring of the manuscript to make the work accessible.

We fully agree and changed both text and figures extensively (see below for details).

The manuscript is rather long and the figures rather large. As far as we understand, the main conclusion is that elasticity must be taken into consideration to explain the shape of gaps within the tissue. Before reaching this conclusion based on the data shown in Fig5, the authors describe at length a capillary model before explaining that this description is not appropriate. Actually, a large part of the data and analysis shown in Fig 1 to 4 becomes irrelevant to the conclusion since they rely on the assumption that the tissue cannot be simply considered as liquid-like.

We draw two main conclusions: (i) gap size, i.e. cell packing density, is not related to adhesion strength; and (ii) elasticity must be considered when modeling adhesion. When rearranging the data, we tried to keep those relevant to one or the other main conclusion but removed those not essential, or not relevant to any of these conclusions. For example, the side length - side number relationship is not relevant for conclusion (ii) but supports and details conclusion (i) and was moved to Fig.1.

Concretely, many considerations (e.g. tensions at gaps deduced from angles, of tissue surface tension, of mean angle comparisons that turn out to not be normally distributed and cannot be compared as such) end being irrelevant for the conclusions.

Tissue surface tensions are empirical, measured data. Contact angles are still valid parameters in both models (as seen from Eqs.5,6). In the new Fig.6 they are used to characterize properties of FN as an adhesion molecule (e.g. explaining why FN does not seem to affect total adhesion strength: adhesion effects are cancelling out). Overall, we tried to make clear now where appropriate how the different observed parameters are related to capillarity-based or the elasto-capillary models,

respectively.

This lack of hierarchy in the presentation of data makes it very difficult for the reader to follow the authors train of thought.

We rearranged the sequence of results and made clear which results are affected by moving to the elasto-capillary model and which are independent. Occasionally we had to refer to conclusions reached only later, in the model, already earlier in the text, which is not elegant but may be helpful.

Similarly, some of the figures are rather large (more than 10 panels) and often contain redundant information (e.g. average value of angle with 2 decimals given below a graph that shows individual measurements as well as the average) or information that may not be essential enough to be shown in a main figure (what is the benefit of showing the brackets, or coloured dots in fig 3D?).

Average values, etc., below graphs, brackets in question have been removed. Coloured dots have been explained in main text and legend. We removed unnecessary and redundant panels: Fig.2A-D; Fig.4C-H; Fig.7E-G; Fig.S1; Fig.S2. We moved panels to supplement: Fig.3A-C; Fig.5A,B, J,K. New panels: Fig.3K,L were derived from and replacing former Fig.5J,K. Remaining panels were rearranged in new Figs. 1-3 and Fig.6. New Fig.S6 was added to show requested data.

At the same time, some data is shown incompletely (e.g. fig 1 gives mean and p values but not the dispersion of the data,

Has been added.

fig 2 no image of control vs morphant aggregates is shown or the raw image used to compute surface tension)

Figs.1A-D illustrated technical details already published elsewhere by us and were deleted to make the figure more succinct. Fig.1C,D showed examples to illustrate how a step in the ADSA procedure looks like. For Cad depletion this was shown e.g. in Ninomiya et al. 2012, J. Cell Sci. 125, 1877-1883, Fig.2D-G.

or incorrectly shown (e.g. fig 5j-k shows the ratio of surface as a function of side length when the relevant information is the distribution given as numbers pasted on the graph,

Former Fig.5J,K has been moved to the supplement (Fig.S5A,B) to show the original data points but the relevant summary data are shown now in the new Fig.3K,L.

control and morphant measurements should be displayed side by side rather than in separated graphs for readers to be able to compare them).

Grey horizontal reference lines have been added to facilitate comparison in Figs1K,L; 6A-F.

Again, the lack of proper hierarchy in the data that is displayed makes it difficult for the reader to follow the relevant information.

Together, the decluttering of figures by removing non-essential panels, the rearrangement of panels and the restructuring of the main text should it make easier to follow the relevant information.

We think that the manuscript would very much benefit from a clearer organization before we could comment on the other smaller issues such as: measurements (mostly on fixed tissue when the gaps are clearly dynamic,

If we would measure angles or gap side lengths in movie frames, this would not be different from measuring these in similarly “frozen” TEM images, except for the much higher resolution in the latter case. A different, exciting approach would of course be to measure individual angles or gap sides evolving over time, at higher than 1 min temporal resolution, together with actin cortex

behavior, to explore concepts like deviations from equilibrium states, elastic energy dissipation, etc., (like what we have done in Parent et al. 2024, Dev. Cell 59, 141-155), but this would amount to a complete, separate paper. Here we were concerned with outlining basic features of a new tissue cohesion model.

often at unknown depth of the contact between cell when the equatorial plane should probably be considered,

In randomly packed tissues, equatorial planes of cells can only be hit for a few random cells in TEM sections or movie focal planes, with most cells or gaps being sectioned at random angles and depths. The effects of random sectioning planes on gap size l and contact angle θ_i has been addressed extensively in our previous publications and respective conclusions and qualifications together with references are given in the Methods section (p.17 bottom). With this, we trust that our conclusions which are essentially based on relative measurements (e.g. gap size l compared in normal and FN-depleted tissues) are not significantly affected.

tension measurements should be given in SI units i.e. N/m and not J/m²),

J is as well a derived SI unit as N , and J/m^2 seems to be commonly used in the context of tissue surface tensions. It is suggestive of tension representing an energy per area, just like e.g, released binding energy per area, B .

statistical (not all data is normally distributed)

Attempting to calculate p using a Mann-Whitney-U-test calculator, we were advised in all cases to use the t-test instead since the data were statistically, for the purpose of hypothesis testing, compatible with a normal distribution (e.g. often we had a very high N). The Mann-Whitney test gave no different conclusions when performed in addition, but nevertheless, in Fig.3G where we decidedly point out the two subpopulations of angles, we add the p value according to the Mann-Whitney test for the combined population in the legend.

or conceptual (seeing how dynamic the gap shape is, how is the cytoskeleton considered here? The cytoskeleton would be equally likely to the PCM to provide elastic contribution to shape the kinks at gaps) considerations

We argue as previously (now last paragraph on p.11) that cortex half-lives and tension decay times are too short for cortex elasticity to be essential in the model. And we added Fig.S6 to show cortical F-actin dynamics at sites of kink formation, with no indication of a direct local effect of the cortex.

We apologize again for the ill-written, confused initial version of the manuscript, and thank the reviewers for their truly constructive criticism.

Second decision letter

MS ID#: dev.204663R1

MS TITLE: Adhesion strength, cell packing density, and cell surface buckling in pericellular matrix-mediated tissue cohesion

AUTHORS: Rudolf Winklbauer; Olivia Luu; Debanjan Barua; Martina Nagel; Yunyun Huang

Dear Dr Winklbauer,

I have now received all the referees' reports on the above manuscript, and have reached a decision. The referees' comments are appended below, or you can access them online: please go to:

As you will see, the referees express considerable interest in your work, but have some significant criticisms and recommend a substantial revision of your manuscript before we can consider publication. If you are able to revise the manuscript along the lines suggested, which may involve further experiments, I will be happy receive a revised version of the manuscript. Your revised paper will be re-reviewed by one or more of the original referees, and acceptance of your manuscript will depend on your addressing satisfactorily the reviewers' major concerns. Please also note that Development will normally permit only one round of major revision. If it would be helpful, you are welcome to contact us to discuss your revision in greater detail. Please send us a point-by-point response indicating your plans for addressing the referees' comments, and we will look over this and provide further guidance.

Please attend to all of the reviewers' comments and ensure that you clearly highlight all changes made in the revised manuscript. Please avoid using 'Tracked changes' in Word files as these are lost in PDF conversion. I should be grateful if you would also provide a point-by-point response detailing how you have dealt with the points raised by the reviewers in the 'Response to Reviewers' box. If you do not agree with any of their criticisms or suggestions please explain clearly why this is so.

Reviewer 1

Through a combination of EM, conventional microscopy and modelling, the paper outlines the roles of pericellular matrix (PCM) in regulating intercellular (interstitial) gaps in development tissue, including partial independence from adhesion strength and counterintuitive effects of high and low PCM abundance.

The paper is much improved. I am satisfied with the science and the revised presentation. However, the text is still not very easy to read and if the authors are able to edit/re-write for clarity and flow, that would be a service to the reader and increase the impact of the paper among non-hyper-specialists. I would also suggest that the Abstract should be a bit more explicit in explaining that the paper supports three pericellular conditions that establish intercellular gaps: sparse PCM with large PCM-poor gaps, moderate PCM when gaps are small and determined by PCM projections from adjacent cells interdigitating, and dense PCM which squeezes out adhesion contacts, creating large gaps but with increased stiffness. This seems to me to be the essence of the paper and should therefore perhaps be stated explicitly in the Abstract.

Reviewer 2

The manuscript from Winklbauer and colleagues is much improved and easier to follow. The experiments and observations are very interesting. The proposed model is useful to push the field in new directions and will trigger exciting discussions and new experiments.

Most of the technical limitations are discussed and taken into account when interpreting the data but there are two instances in which I think these should be stated more explicitly for readers who may not be entirely familiar with the fields or with the approaches used in the manuscript. Also, some terms such as "adhesion strength" are often used in ways that may be confusing.

First, the assumption that tissue surface tension measurements can be used to measure adhesion strength is rather bold now that we understand better how cell-cell contacts form and detach (Maitre 2012, Engl 2014, Kashkooli 2021 or Arora 2025; reviewed in Maitre and Heisenberg 2013, Lecuit and Yap 2015 or Campas 2024). Compression of explants/aggregates/spheroids will be resisted by adhesion strength but also by cell stiffness or the production other stiff components such the PCM for example. How can the authors rule out that FN MO show a similar tissue surface tension to control while having lower adhesion strength but stiffer cells or PCM for example? This assumption needs to be better justified and explicitly mentioned in the manuscript. Also the term "adhesion strength" is used repeatedly throughout the manuscript, which discusses both cell-cell adhesion and cell-ECM (or PCM) adhesion. I suspect the term "adhesion strength" mostly refers to cell-cell adhesion but I imagine it could be difficult to guess or disorienting for future readers. Using a different term may be advisable.

Second, another important assumption made by the authors in their interpretation is the interpolation of their 2D measurements on fixed samples to a dynamic 3D tissue. Many of the very interesting measurements of contact angles and kinks at contact gaps are done in 2D with little information about the orientation of the contacts that are being measured. Did the authors only measure cells that were captured in their equatorial plane? Measuring contact geometrical parameters on a tilted contact would certainly affect the quality of the measurement. Also, how are the contacts chosen? What software is being used to measure the angles? Moreover, live imaging provided by the authors reveals dynamic changes of the contact angles, gap size and kinks over short time of a few minutes. This corresponds very much to the time scales of rearrangement of the actomyosin cytoskeleton, as reported by the authors and as supported by a very large body of literature. Yet the authors dismiss the role of actomyosin contractility stating that a fast turnover of 1-2 min would be "significantly faster than in the PCM" (2-4 min according to Fig4-5). I disagree with this statement. Contact shape fluctuations resulting from actomyosin fluctuations do not necessarily need actomyosin accumulations directly at the site of deformation since actomyosin can move intercellular fluid and deform gaps at a distance as observed for example in xenopus gastrula (Park 2015) or mouse embryos (Dumortier 2019 and Schliffka 2024). The alternative hypothesis proposed by the authors that PCM turnover may explain the dynamic changes in gap shape is interesting but would need to be supported by a characterization of the dynamic turnover of the PCM. I would encourage the authors in revising their interpretations and discussions of these results.

Finally, the authors often compare the control and FN MO conditions but in their figures, the measurements for control and FN MO conditions are often in separate graphs making it difficult to appreciate and prevents the authors from including statistical analysis that would support their claims. Control and FN MO measurements should be merged and tested in Fig 1K-L, 3K-L, 6A-F.

Can the authors add a schematic and/or picture of surface tension measurement before the panels Fig 2A-C?

In Fig 2D", the authors depict a gap within tissues but this gap is very different from the ones seen in the EM images of the study. This schematic shows a gap located between 2 cells only, forming a pocket, while the gaps that are characterized later on are bounded by more than 2 cells. This schematic gives the incorrect idea of the topology of the tissue and should be amended.

Second revision

Author response to reviewers' comments

Response to reviewers

We thank the reviewers for pointing out remaining clarity issues to further improve the manuscript.

Reviewer 1: Through a combination of EM, conventional microscopy and modelling, the paper outlines the roles of pericellular matrix (PCM) in regulating intercellular (interstitial) gaps in development tissue, including partial independence from adhesion strength and counterintuitive effects of high and low PCM abundance.

The paper is much improved. I am satisfied with the science and the revised presentation. However, the text is still not very easy to read and if the authors are able to edit/re-write for clarity and flow, that would be a service to the reader and increase the impact of the paper among non-hyper-specialists.

We agree, clarity and readability of the text should be improved. We amended all sections of the manuscript, rearranged some paragraphs, moved technical remarks to the figure legends, removed non-essential, confusing statements, and added clarifying comments and two explanatory supplementary figures.

I would also suggest that the Abstract should be a bit more explicit in explaining that the paper supports three pericellular conditions that establish intercellular gaps: sparse PCM with large PCM-poor gaps, moderate PCM when gaps are small and determined by PCM projections from adjacent cells interdigitating, and dense PCM which squeezes out adhesion contacts, creating large gaps but with increased stiffness. This seems to me to be the essence of the paper and should therefore perhaps be stated explicitly in the Abstract.

In the model section, two theoretical mechanisms are discussed: low PCM abundance leading to “naked” gaps, and high abundance involving PCM compression and ensuing effects. In both cases, interdigitation of PCM projections would be involved (see Fig.7A), so we do not consider this a separate mechanism. We emphasize in the 1st paragraph of the Discussion that our paper supports the second (high PCM) mechanism, with no evidence for the first, theoretical possibility (although this could be different in a different tissue). We think the Abstract in its present form reflects this situation. It focusses on the probably more consequential empirical findings: independence of adhesion strength and cell packing density, and solid-like behavior of cell surfaces. The relevance of these features for tissue architecture is mentioned in the last Discussion paragraph, and linking them into a model draws attention to the role of the PCM in adhesive contact and gap formation.

Reviewer 2: The manuscript from Winklbauer and colleagues is much improved and easier to follow.

The experiments and observations are very interesting. The proposed model is useful to push the field in new directions and will trigger exciting discussions and new experiments.

Thank you for these encouraging remarks which express all we hope for.

Most of the technical limitations are discussed and taken into account when interpreting the data but there are two instances in which I think these should be stated more explicitly for readers who may not be entirely familiar with the fields or with the approaches used in the manuscript. Also, some terms such as “adhesion strength” are often used in ways that may be confusing.

First, the assumption that tissue surface tension measurements can be used to measure adhesion strength is rather bold now that we understand better how cell-cell contacts form and detach (Maitre 2012, Engl 2014, Kashkooli 2021 or Arora 2025; reviewed in Maitre and Heisenberg 2013, Lecuit and Yap 2015 or Campas 2024). Compression of explants/aggregates/spheroids will be resisted by adhesion strength but also by cell stiffness or the production other stiff components such the PCM for example.

We explain now that one possible and valid measure of adhesion strength is “surface tension” (extensively discussed in Winklbauer, 2015). It expresses the difference of free energy at contact and non-contact surfaces at equilibrium, i.e. after cells have completed attachment or separation processes. This avoids the need to include the complicated dissipative processes that are important in adhesion kinetics, e.g. during the pulling apart of two cells or of an aggregate, and that depend on additional parameters like separation velocities. We now mention these alternatives, valuable in their own right, together with respective references in the Results (p.6, 2nd paragraph). Our short definition of adhesion strength in the Introduction (p.4, 1st paragraph) also implies that surface tension is just one way of measuring it, but that conveniently, it is directly related to the tension equilibria in our diagrams and equations.

How can the authors rule out that FN MO show a similar tissue surface tension to control while having lower adhesion strength but stiffer cells or PCM for example? This assumption needs to be better justified and explicitly mentioned in the manuscript.

This issue is explored between p.11, bottom line, and p.13, top paragraph. Since we define adhesion strength here as what is measured as tissue surface tension σ , one cannot be lower than the other. In fact, σ represents the reversible work per area, needed to create free cell surface area from contact area, regardless of whether it contains an elastic component (Eqn 5, 6) or not (Eqn 1, 2). And regardless of the speed of the process! What changes is the distribution of energies between its different components. For example, the binding energy released, B , contributes to adhesion tension Γ and to elasticity in the first case, but to Γ only in the second. We added a clarifying remark at the end of this section (p.13, top).

Also the term "adhesion strength" is used repeatedly throughout the manuscript, which discusses both cell-cell adhesion and cell-ECM (or PCM) adhesion. I suspect the term "adhesion strength" mostly refers to cell-cell adhesion but I imagine it could be difficult to guess or disorienting for future readers. Using a different term may be advisable.

We defined now "adhesion strength" in more detail and use the term more consistently (replacing e.g. "adhesiveness" which we had occasionally used for variety of expression when we had meant "adhesion strength"; but still use "relative adhesiveness" for the dimensionless form). We always mean cell-cell adhesion which, being mediated by PCM-PCM interaction, we equate with PCM-PCM adhesion (the PCM being the cell coat, p.4, 3rd paragraph). We do not discuss the cell-PCM connection but mention now briefly that we consider it as always sufficiently strong in the present context (p.10 middle paragraph).

Second, another important assumption made by the authors in their interpretation is the interpolation of their 2D measurements on fixed samples to a dynamic 3D tissue. Many of the very interesting measurements of contact angles and kinks at contact gaps are done in 2D with little information about the orientation of the contacts that are being measured. Did the authors only measure cells that were captured in their equatorial plane? Measuring contact geometrical parameters on a tilted contact would certainly affect the quality of the measurement.

We had shortly mentioned this problem, indeed a source of systematic error, in the Materials and Methods together with respective references to our earlier work. We expanded this part now and explain the issue in more detail (p.18, top), and hope that the arguments can be understood now without consulting the references. As a main conclusion, we mention that in the present manuscript we refer only to relative changes of lengths (e.g. upon FN depletion), which should not be affected by proportional deviations from true lengths. Average absolute angle values are relied on (e.g. the kink angle) but for angles, we noted previously and mention now that the distribution width, not the average, is affected by random sectioning planes.

Also, how are the contacts chosen? What software is being used to measure the angles?

We mention in Material and Methods that all contacts between interstitial gaps in a TEM image were measured, and added that Axiovision software was used for angle and length measurements.

Moreover, live imaging provided by the authors reveals dynamic changes of the contact angles, gap size and kinks over short time of a few minutes. This corresponds very much to the time scales of rearrangement of the actomyosin cytoskeleton, as reported by the authors and as supported by a very large body of literature. Yet the authors dismiss the role of actomyosin contractility stating that a fast turnover of 1-2 min would be "significantly faster than in the PCM" (2-4 min according to Fig4-5). I disagree with this statement. Contact shape fluctuations resulting from actomyosin fluctuations do not necessarily need actomyosin accumulations directly at the site of deformation since actomyosin can move intercellular fluid and deform gaps at a distance as observed for example in *xenopus gastrula* (Park 2015) or mouse embryos (Dumortier 2019 and Schliffka 2024). The alternative hypothesis proposed by the authors that PCM turnover may explain the dynamic changes in gap shape is interesting but would need to be supported by a characterization of the dynamic turnover of the PCM. I would encourage the authors in revising their interpretations and discussions of these results.

We completely agree with the reviewer's notion of how dynamic gap shapes are connected to the dynamic actomyosin cytoskeleton. We think that cortex contractility fluctuations drive the shape change transitions in gaps and contacts, which would explain the similar time scales. However, we discuss in addition the possibility that the cortex also contributes to the surface stiffness, which in turn determines not the temporal pattern of surface shape changes, but the actual shapes, e.g. being kinked or sinusoidal or bulging. For this role, we think the actomyosin cortex is not well suited due to its mechanical characteristics (turnover time etc.). Instead, we propose that the PCM under compression stiffens the cell surface, assuming (or: predicting) that PCM turnover is sufficiently slow to allow for elastic stresses to be maintained for sufficiently long times (minutes). In the previous version of the paper, this argument had been scrambled and spread out over

different pages. We now concentrate it in the last paragraph of the Model section, and hope in this way to clarify the point. We also mention that special cortex features are not seen at kinks.

Finally, the authors often compare the control and FN MO conditions but in their figures, the measurements for control and FN MO conditions are often in separate graphs making it difficult to appreciate and prevents the authors from including statistical analysis that would support their claims. Control and FN MO measurements should be merged and tested in Fig 1K-L, 3K-L, 6A-F.

We added significance levels comparing control and FNMO values in Fig. 1K,L. In Fig. 3K,L no comparison is made between treatments, no relevant differences or equalities are claimed between control and FNMO. In Fig. 6A-F orientation is indeed difficult. We added significance levels comparing control and FNMO in Fig. 6A,B to support our respective claims. But it is difficult to see how in Fig. 6C-F, the complex data clouds in the panels could be merged. To help with orientation, we had introduced colored data points, but in addition we connect now with arrows the panels C-F with the data in A-B from which they originate.

Can the authors add a schematic and/or picture of surface tension measurement before the panels Fig 2A-C?

To keep the main section including the figures short and simple, we added a schematic of the tissue surface measurement procedure as supplementary Figure S2. We refer to it also in the Materials and Methods section, where details of the procedure and references can be found.

In Fig 2D", the authors depict a gap within tissues but this gap is very different from the ones seen in the EM images of the study. This schematic shows a gap located between 2 cells only, forming a pocket, while the gaps that are characterized later on are bounded by more than 2 cells. This schematic gives the incorrect idea of the topology of the tissue and should be amended.

This touches on an interesting point, which we had avoided though to keep the text simple. We add now a supplementary Fig. S3 where we extend for the interested reader our argument from the special case of 2-sided gaps to irregular 3-sided gaps (from this it should be easy to see how it applies to all n). In the Results section, we re-worded and rearranged the argument (p,7 top paragraph) and used e.g. "generalized interstitial space" instead of "2-sided gaps", while referring to the new Fig.S3.

Third decision letter

MS ID#: dev.204663R2

MS TITLE: Adhesion strength, cell packing density, and cell surface buckling in pericellular matrix-mediated tissue cohesion

AUTHORS: Rudolf Winklbauer; Olivia Luu; Debanjan Barua; Martina Nagel; Yunyun Huang
ARTICLE TYPE: Research Article

Dear Dr Winklbauer,

I am happy to tell you that your manuscript has been accepted for publication in Development, pending our standard publication integrity checks.

Reviewer 1

I am satisfied with the additional revisions.

Reviewer 2

I thank the authors for their responses and modifications. I think the manuscript has been further improved but I would encourage them to think once again about those 2 aspects:

-I still think the Fig 2D" is misleading readers into thinking that gaps are pockets contained by 2 cells only. This also raises the question of the authors actually know whether a given gap is 2 sided or 3- or multiple-sided without having full 3D imaging.

-"We explain now that one possible and valid measure of adhesion strength is "surface tension" (extensively discussed in Winklbauer, 2015). It expresses the difference of free energy at contact and non-contact surfaces at equilibrium, i.e. after cells have completed attachment or separation processes. "

One major assumption of making an analogy between cells and soap bubbles to describe adhesion is the one of reversibility. We now know fairly well that the formation and separation of cell-cell contacts are very different (a major difference being that it is unclear how many adhesion molecules actually detach their trans-binding (Maitre 2012, Rozema 2023, Arora 2025)). Using the term adhesion strength to describe the difference in free energy of a contact for BOTH its formation and destruction is not in agreement with recent literature. I would encourage the authors to clarify this situation by discussing it more explicitly. This will benefit the readers who may otherwise be misled.